# Brain-Derived Neurotrophic Factor, Nociception, and Pain

**DOI:** 10.3390/biom14050539

**Published:** 2024-04-30

**Authors:** Adalberto Merighi

**Affiliations:** Department of Veterinary Sciences, University of Turin, 10095 Turin, Italy; adalberto.merighi@unito.it; Tel.: +39-0116709118

**Keywords:** BDNF, nociception, pain, neurons, glia, presynaptic mechanisms, postsynaptic mechanisms, LTP, neuropeptides

## Abstract

This article examines the involvement of the brain-derived neurotrophic factor (BDNF) in the control of nociception and pain. BDNF, a neurotrophin known for its essential role in neuronal survival and plasticity, has garnered significant attention for its potential implications as a modulator of synaptic transmission. This comprehensive review aims to provide insights into the multifaceted interactions between BDNF and pain pathways, encompassing both physiological and pathological pain conditions. I delve into the molecular mechanisms underlying BDNF’s involvement in pain processing and discuss potential therapeutic applications of BDNF and its mimetics in managing pain. Furthermore, I highlight recent advancements and challenges in translating BDNF-related research into clinical practice.

## 1. Introduction

### 1.1. A Brief Overview of BDNF Functions in the Normal and Pathologic Nervous System and the Cellular Processing of BDNF

Brain-derived neurotrophic factor (BDNF) is a pivotal molecule in the field of neuroscience and neurobiology, playing a crucial role in neuronal differentiation, development, survival, and maintenance; it also acts as a neurotransmitter and intervenes in the plasticity of the nervous system [1]. However, the role of BDNF extends beyond the realm of basic neuroscience and influences various aspects of brain health and disease. After the initially discovered functions of BDNF on neuronal differentiation, survival, and growth, several other effects were revealed, such as intervention in synaptic plasticity [2,3], neuroprotection [4,5], neurodegeneration [1], and the control of mood disorders [2]. Thus, most of the originally discovered roles of this neurotrophin shape the cellular and functional organization of the normal brain, but more recent findings have converged to demonstrate that dysregulation of BDNF is implicated in a range of neurological disorders, including Alzheimer’s disease [4,5], Parkinson’s disease [6], Huntington’s disease [7,8], and amyotrophic lateral sclerosis (ALS) [9,10] and that low BDNF levels are associated with depression [11] and anxiety [12]. As I will discuss in this review, BDNF is today considered a neurotransmitter that intervenes in the modulation of pain. Pain may be considered physiological, but it is also detrimental to the organism, thus the role of BDNF has become progressively more important, with the accumulation of data converging to demonstrate its widespread distribution and functions in pain pathways.

BDNF is a member of the neurotrophin family of proteins, which includes nerve growth factor (NGF), neurotrophin-3 (NT-3), and neurotrophin-4/5 (NT-4/5). Much of the work on neuronal BDNF synthesis, secretion, and release as well as interactions with its high-affinity receptor, tropomyosin-related receptor kinase B (trkB), was originally carried out in the hippocampus, neocortex, and basal forebrain, which are pivotal to higher brain functions. Yet the synthesis and processing of neurotrophins follow the same or similar pathways in the peripheral and central neurons processing nociceptive and pain information.

BDNF is primarily synthesized as a precursor protein known as proBDNF (Figure 1). This precursor molecule is then proteolytically cleaved to generate mature BDNF (In the following sections, I will use simply BDNF to indicate the mature BDNF protein. Only in this section, I will use the definition of mature BDNF to distinguish the mature form of the protein from its precursors), which is the biologically active form [13], although proBDNF has been recognized to also play a role in regulating neuronal activity [14].

The mature BDNF protein consists of 247 amino acids and forms a homodimer. Each monomer consists of a long pro domain, followed by a mature BDNF domain. The pro-domain is cleaved during secretion to release the biologically active mature BDNF. Cleavage is essential for BDNF’s proper functioning. In initial studies, it was thought that neurons only release the mature form of the neurotrophin, but, it was later demonstrated that both proBDNF and mature BDNF can be released by neurons [14]. Once released, BDNF and proBDNF can initiate a series of intracellular signaling pathways that play crucial roles in the numerous functions of the neurotrophin [13]. Notably, however, proBDNF and mature BDNF activate different receptors to produce their cellular effects. Specifically, proBDNF activates the low-affinity p75 pan neurotrophin receptor (p75^NTR^) rather than trkB [14] (Figure 1). Inhibition of trkB blocks BDNF signaling, whereas blocking p75^NTR^ prevents the signaling of proBDNF. Among the several functions of BDNF or proBDNF upon receptor binding, the regulation of synaptic activity is of relevance to the present discussion. It is today widely accepted that mature BDNF can induce long-term potentiation (LTP), whereas proBDNF sustains long-term depression (LTD) at synapses following specific patterns of activity [15].

Sustained by BDNF/trkB, LTP is related to the plasticity and subsequent sensitization of the synapse. To induce LTP, BDNF has the potential to directly affect excitatory neurons both pre- and postsynaptically (Figure 2). Additionally, it can alter the balance between excitation and inhibition by inhibiting the GABAergic neurons [16]. For LTP to rise, both the pre- and postsynaptic neurons must be active simultaneously. This is because the postsynaptic neuron needs to be depolarized when glutamate is released from the presynaptic bouton to completely remove the Mg^2+^ block of N-methyl-d-aspartate (NMDA) receptors (NMDARs). BDNF-induced LTP involves a series of complex current-voltage relationships among NMDARs and α-amino-3-hydroxy-5-methyl-4-isoxazole propionic acid (AMPA) receptors (AMPARs) [16]. Both receptors are ionotropic and allow the passage of Na^+^ and K^+^ ions, resulting in a significant influx of the former and a little efflux of the latter, thus causing the postsynaptic neuron to depolarize. When depolarization and glutamate binding occur at the same time, it leads to the maximum influx of Ca^2+^ ions through NMDARs. This Ca^2+^ influx then triggers several intracellular signaling cascades, which ultimately cause changes in synaptic efficiency (Figure 2). NMDARs mostly operate at the postsynaptic membrane, but they have also been observed on presynaptic boutons, where they play a role in controlling the release of fast-acting transmitters [17]. The activation of these presynaptic receptors can occur through two mechanisms: homosynaptic modulation, which involves significant release from the bouton on which they are situated, and heterosynaptic modulation, which involves the release of glutamate by nearby synapses. The impact of this activation on subsequent release is contingent upon the specific type of synapse in question.

Differently from LTP, LTD can be stimulated by repeated activation of the presynaptic neuron at low frequencies in the absence of postsynaptic activity. Due to the substantial driving force for Ca^2+^ entry in a resting neuron and the incomplete blockage of NMDARs by Mg^2+^ even at resting potentials, a significant influx of Ca^2+^ occurs in response to low-frequency synaptic stimulation. The repeated occurrence of this lower NMDAR-dependent Ca^2+^ influx is likely what drives the generation of LTD [15].

Since NMDAR-dependent calcium influx triggers both LTP and LTD, the cell needs a mechanism to determine whether to strengthen or weaken a synaptic connection. Today, it is well acknowledged that moderate activation of NMDARs, resulting in a moderate increase in postsynaptic Ca^2+^, is ideal for initiating LTD. On the other hand, considerably stronger activation of NMDARs, leading to a much bigger increase in postsynaptic Ca^2+^, is necessary to cause LTP. Weak activity of the presynaptic neuron leads to modest depolarization and Ca^2+^ influx through NMDARs. This activates cellular phosphatases that dephosphorylate AMPARs, thus promoting receptor endocytosis and reinforcing LTD. Remarkably, AMPAR endocytosis can also be sustained by proBDNF after binding of p75^NTR^ (Figure 1). On the other hand, strong activity matching strong depolarization, as well as mature BDNF binding to trkB, triggers LTP and the subsequent activation of calcium/calmodulin-dependent protein kinase II (CaMKII), AMPAR phosphorylation, and exocytosis (Figure 1).

### 1.2. Pain as a Complex and Multifaceted Sensory Experience

Pain is a universal and complex phenomenon in the animal world that has intrigued scientists, clinicians, and philosophers for centuries. It transcends mere sensory perception and encompasses a multidimensional experience that involves not only the discernment of noxious stimuli but also emotional, cognitive, and behavioral responses. This multifaceted nature of pain makes it a subject of great interest and study in various fields, including neuroscience, psychology, and clinical medicine.

Pain can be broadly defined as an unpleasant sensory and emotional experience associated with actual or potential tissue damage [18]. It serves as a vital protective mechanism, alerting individuals to harmful stimuli and prompting them to take action to prevent further injury. However, pain is far from being a simple “alarm system” for the body; instead, it exhibits complexity at multiple levels.

At its core, pain begins with nociception, the detection of noxious or potentially harmful stimuli, and its transduction into electric signals by specialized receptors known as nociceptors [19]. These receptors lie in the peripheral nervous system (PNS) and are scattered throughout the body, from the skin to the internal organs [20]. When stimulated, nociceptors initiate a cascade of events that lead to the transmission of pain signals from the spinal nerves and some cranial nerves that possess somatic and visceral sensory fibers to the central nervous system (CNS) [21]. Beyond nociception, pain involves emotional and affective components. The brain processes these components of pain in brain regions associated with emotions, such as the amygdala (AMYG) and anterior cingulate cortex (ACC) [22]. This emotional aspect of pain gives rise to the subjective experience of suffering, distress, and fear often associated with painful stimuli. Emotional responses to pain can vary greatly among individuals and are influenced by factors like previous experiences, psychological state, and cultural background. Cognition plays a critical role in shaping the perception of pain. Cognitive factors, such as attention, expectation, and belief, can significantly influence how individuals perceive and respond to painful stimuli. Therefore, cognitive strategies, such as distraction or cognitive reappraisal, can either exacerbate or mitigate the experience of pain. Sociocultural and contextual factors also influence pain.

**Figure 1 biomolecules-14-00539-f001:**
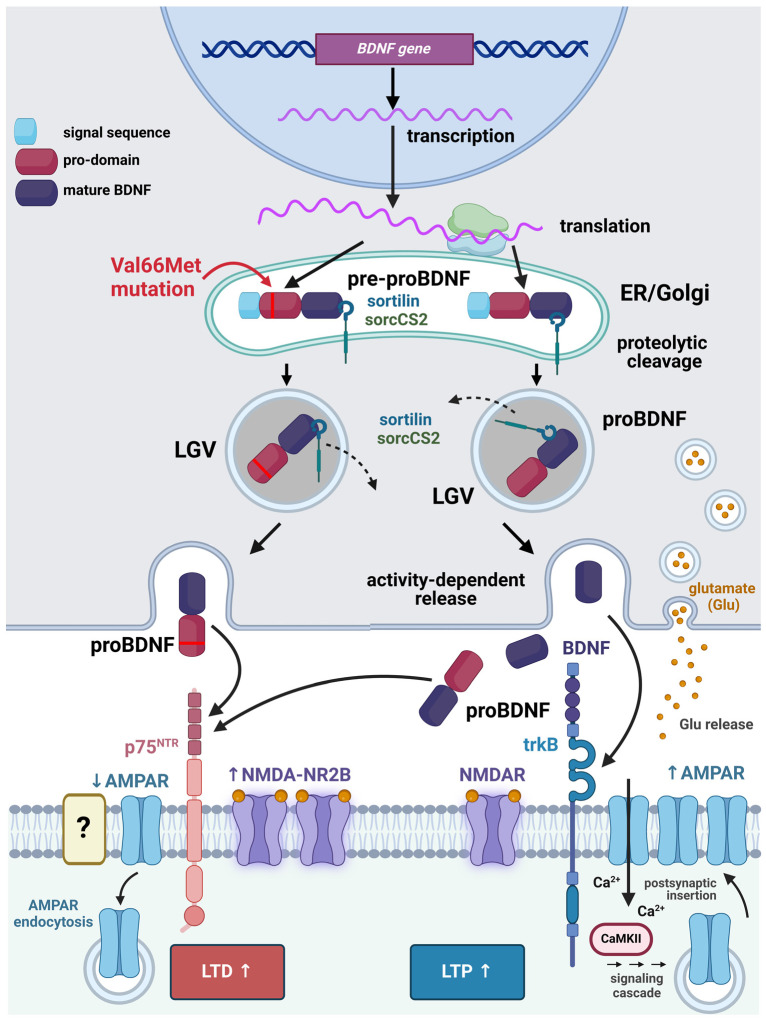
Synthesis, secretion, and effects of BDNF and its pro-peptide (proBDNF) on plasticity at the primary sensory neuron (PSN)/secondary sensory neuron (SSN) synapse. PSNs use glutamate as a fast-acting neurotransmitter and some slow-acting neurotransmitters, among which is BDNF [23]. BDNF is first produced as a pre-pro-protein precursor (BDNF pre-pro-protein), which consists of a signal sequence (light blue), a pro-domain (magenta), and the mature BDNF peptide (cobalt). Upon entering the endoplasmic reticulum (ER), the signal sequence is promptly removed to produce the BDNF pro-protein (proBDNF), which is then transported to the trans-Golgi network (TGN). In the TGN, proBDNF binds to sortilin or sortilin-related VPS10 domain-containing receptor 2 (sorCS2) to be sorted in dense-core large granular vesicles (LGVs). In mature LGVs, proBDNF detaches from sortilin and becomes ready for release. The Val66Met BDNF polymorphism, represented by a thick red line, influences the release of BDNF in response to activity. The mature BDNF peptide is produced through proteolytic cleavage of proBDNF within presynaptic LGVs and then released by either the constitutive (not represented) or regulated activity-dependent pathway. BDNF and its precursor protein respectively promote LTP and LTD. For schematization, LTD and LTP are shown on the left and right sides, respectively, of the same postsynaptic neuron, but they can occur independently as explained in the main text. The question mark indicates that the cellular pathways activated by proBDNF may involve other not yet identified membrane receptors. Abbreviations: AMPAR = α-amino-3-hydroxy-5-methyl-4-isoxazole propionic acid receptor; BDNF = brain-derived neurotrophic factor; CaMKII = calcium/calmodulin-stimulated protein kinase II; ER = endoplasmic reticulum; LGV = large granular vesicles; LTD = long-term depression; LTP = long-term potentiation; p75^NTR^ = low-affinity p75 pan neurotrophin receptor; proBDNF = BDNF pro-protein; PSN = primary sensory neuron; sorCS2 = sortilin-related VPS10 domain-containing receptor 2; SSN = secondary sensory neuron; TGN = trans-Golgi network; trkB = tropomyosin receptor kinase B; ↑ = increase; ↓ = reduction. Created with BioRender.com.

According to the International Association for the Study of Pain (IASP), there are several types of pain (https://www.iasp-pain.org/resources/terminology/ accessed on 29 April 2024) [18]. Nociceptive pain is pain that arises from non-neural tissues and is a consequence of the activation of nociceptors. Neuropathic pain is a type of pain caused by a lesion or disease of the somatosensory pathways. It can be further classified as central or peripheral according to the division of the somatosensory system that is lesioned. Nociplastic pain arises from altered nociception, despite no clear evidence of actual or threatened tissue damage activating peripheral nociceptors or evidence of disease or lesion of the somatosensory system causing the pain. There are also differences between acute and chronic pain [24]. Acute pain is self-limited, is triggered by a particular illness or injury, and has a protective function. On the other hand, chronic pain might be seen as a medical condition. It is discomfort that, whether connected to a sickness or injury, lasts longer than the typical recovery period. Chronic pain lacks a biological purpose, can have psychological causes, and has no obvious endpoint. Chronic pain conditions, characterized by persistent pain lasting beyond the expected healing period, represent a particularly challenging aspect of pain.

The anatomical arrangement of pain circuits is summarized in Box 1 to describe the general organization of somatic and visceral pain pathways. The Figure in the box summarizes the data on the localization of BDNF in these pathways. More comprehensive accounts can be found, e.g., in refs. [21,25].

## 2. Role of BDNF in Nociception and Pain: Insights and Mechanisms

Under basal conditions, BDNF is synthesized by various types of neurons and glia within pain pathways. Noxious stimuli can trigger the production and release of BDNF by these cells and/or upregulate BDNF synthesis and release. Thus, BDNF is involved in the induction of a form of synaptic plasticity, leading to an increase in the responsiveness of peripheral nociceptors to nociceptive stimuli. This sensitization process can take place at the level of peripheral nerve endings (peripheral sensitization) or central neurons (central sensitization) [26,27]—see Section 3 BDNF and Neuronal Sensitization. Inflammatory processes are often accompanied by an increased release of BDNF as part of a complex network of signaling cascades. Inflammation-induced BDNF release can have both neuroprotective and detrimental effects, depending on the context and duration of the inflammation [28]. Similarly, lesions of the peripheral nerves leading to neuropathic pain may be accompanied by an upregulation of BDNF and the activation of the BDNF/trkB signaling cascade [29].

In the following subsections, I will review the state-of-art information on the localization of BDNF in cells of the pain pathways.

### 2.1. BDNF in Pain Pathways Neurons

The localization of the BDNF protein in neurons of the somatic and visceral pathways is summarized in Table 1 and Table 2, respectively. It is also shown in graphical form in Box 1. One very important issue to be considered when dealing with BDNF localization studies is that BDNF, differently from other neurotrophic factors, can be anterogradely transported to targets [30,31], this being fundamental to explaining its messenger role in the modulation of synaptic activity [25]. From a different perspective, the anterograde transport of BDNF explains why in certain areas of the CNS, it has been possible to detect the BDNF mRNA in the partial or total absence of protein immunostaining. In addition to BDNF, it is worth noting that pro-BDNF can also be anterogradely transported [32]. Central neurons in the cerebral cortex, parabrachial nucleus (PBN), hippocampus, and locus coeruleus have been, e.g., discovered to synthesize and transport BDNF anterogradely [30,33,34]. Regarding peripheral neurons, peptidergic small- to medium-sized dark neurons in DRGs produce and transport BDNF to their central terminals in the dorsal horn of the spinal cord [35,36].

The overall frame should be completed considering several intricate regulatory processes contributing to BDNF functional flexibility. These processes comprise controlling the movement of different BDNF transcripts within the cells and managing the modification and release of BDNF after protein synthesis. Although they are very important for understanding the context-dependent mechanisms of BDNF signaling [37], they will be mentioned only if directly relevant to the topic of this paper.

#### 2.1.1. Primary Sensory Neurons

As reported in Table 1 and Table 2 and shown in Box 1, PSNs located in different ganglia of the PNS express BDNF under basal conditions and/or under circumstances of altered pain perception. One of the first studies on the localization of BDNF in PSNs reported that about 25% of small- to medium-sized neurons in the rat lumbar DRGs are immunoreactive for the neurotrophin [38]. Subsequently, the same authors have localized BDNF and its mRNA in certain cranial ganglia containing special PSNs (vestibulocochlear and geniculate ganglia that are not relevant to the present discussion), the nodose ganglion (NG), and the trigeminal ganglion (TG) [39]. The neurotrophin was then localized to the jugular ganglion (JG) [40], and subsequent studies (reported in Table 1 and Table 2) were confirmative of these observations. Thus, physiologically, BDNF is expressed by small- to medium-sized PSNs of the DRGs and cranial ganglia containing general somatic or visceral PSNs, except for the superior ganglion of the glossopharyngeal nerve, for which data are not available. The anatomy, function, and neuronal diversity of DRGs have long since been extensively investigated [41,42], and more recently, somatosensory neuron types have been defined with single-cell transcriptomics [43]. The neuronal diversification of cranial sensory ganglia is known in much less detail and has been recently reviewed [44]. The TG houses the PSNs of cranial nerve V, which are accountable for sensing and relaying general somatic stimuli from the skin and deep tissues of the head as well as visceral stimuli from intracranial blood vessels. As a result, the TG is seen as reasonably comparable to the DRGs whose PSNs convey the same types of stimuli from the remaining parts of the body.

DRGs consist of proprioceptive, mechanoceptive, and nociceptive neurons that transmit distinct characteristics of somatosensation. A major difference in the PSN composition between the TG and DRGs is that the former lacks proprioceptive neurons that are instead located within the CNS in the trigeminal mesencephalic nucleus and provide proprioceptive fibers to the masticatory muscles [45], leaving somewhat of an enigma regarding the facial muscles proprioception [46]. DRG proprioceptors and mechanoreceptors are characterized by their large soma and medium- to large-diameter myelinated axons, specifically Aα and Aβ fibers, respectively, in terms of their morphology. Nevertheless, a certain group of neurons that are responsible for transmitting light touch sensations and expressing emotional components of mild touch, known as C- low-threshold mechanoreceptors (C-LTMRs), give rise to unmyelinated axons. These neurons also regulate the transmission of nociceptive stimuli [42,47]. They appear to originate from precursor cells differentiating into nociceptors, which are the third type of PSNs in DRGs. The nociceptive lineage consists of thermoreceptors that detect harmless changes in temperature, pruriceptors that detect itch, and nociceptors that detect harmful thermal, mechanical, and chemical stimuli [48]. Most nociceptive neurons are thus polymodal, meaning that they may respond to a wide range of stimuli. Nociceptors constitute the chief neuronal subtype in both the TG and DRGs. Neurons belonging to the nociceptive lineage are present in the JG, NG, and petrosal ganglion (PG) as well [49]. Nociceptive neurons, in contrast to proprioceptors and mechanoreceptors, are characterized by their small size and possess axon fibers that are either mildly myelinated (Aδ) or unmyelinated (C) [50,51]. The nociceptors can be further classified into peptidergic and non-peptidergic types. The former class provides sensory innervation to the skin as well as deep tissues, including bones and viscera. On the other hand, the latter specifically innervates the epidermis [48]. Peptidergic nociceptors release neuropeptides such as substance P, somatostatin, or calcitonin gene-related peptide (CGRP) [52,53]. Under basal conditions, expression of BDNF has been reported in peptidergic nociceptors containing substance P and CGRP [54,55,56,57] and expressing the transient receptor potential vanilloid receptor 1 (TRPV1) [53,58]. Even under basal conditions, there is another subpopulation of DRG neurons expressing BDNF that is not peptidergic and remains to be characterized in full [57]. These neurons are larger and likely immunoreactive to the RT97 antibody that recognizes the phosphorylated form of neurofilament protein NF200 [50]. Alterations in BDNF immunoreactivity in the L4 and L5 rat DRGs are observed in experimental models of neuropathic pain [59]. Chronic constriction injury (CCI) damage to the sciatic nerve causes a substantial ipsilateral rise in the proportion of small, medium, and large BDNF-immunoreactive neurons. After spinal nerve ligation (SNL), there is instead a considerable rise in the proportion of BDNF-immunoreactive large neurons and a notable drop in the proportion of small BDNF-immunoreactive neurons in the L5 DRG on the same side. Conversely, there is a marked increase in the proportion of BDNF-immunoreactive neurons of all sizes in the L4 DRG. Injection of complete Freund adjuvant (CFA) into the hind paws of rats causes increased sensitivity to mechanical pain and significant elevations in levels of tumor necrosis α (TNFα) in the inflamed tissues accompanied by increases in BDNF, trkB, CGRP, and TRPV1 in the DRGs [60]. Although data on the distribution of BDNF immunoreactivity according to neuronal size were not provided by the authors, an inspection of their images suggests that the increase in BDNF signal occurs in all classes of DGR neurons (small, medium, and large). In another paper in which CFA was used to induce peripheral inflammation, BDNF immunoreactivity increases ipsilaterally in L5 DRG neurons of different sizes [61]. In a different model of pathologic pain (bone cancer), up-regulation of BDNF was instead localized to small but not medium or large neurons [62].

Like those in DRGs, PSNs in TG can be classified based on histological, neurochemical, and functional features [63]. As in DRGs, the peptides CGRP and substance P are commonly colocalized in small-size TG nociceptors. BDNF immunoreactivity is detected in a large subpopulation of TG neurons. Positive neurons are mostly small- or medium-sized, and substantial fractions of these neurons are also stained for CGRP and TRPV1 [64,65].

The proximal ganglia of cranial nerves IX and X (JG) contain specialized somatic sensory neurons that chiefly supply nerve fibers to the external auditory meatus, the dura mater, and the pharyngeal region. Although the neuronal diversity of these ganglia remains uncharacterized, single-cell RNA sequencing investigations have demonstrated a significant resemblance between JG neurons and DRG neurons [49,66,67]. In the JG, about 50% of neurons are BDNF-immunoreactive, and large percentages of these neurons are also immunoreactive for CGRP or TRPV1 [40].

As mentioned previously, the PG and NG are the distal ganglia of cranial nerves IX and X, respectively. The PG harbors the visceral sensory neurons responsible for innervating some respiratory system organs and transmitting taste information from the posterior portion of the tongue. The NG neurons provide sensory innervation to the pharynx area, thoracic organs, and a large portion of the digestive tract. They transmit information regarding stretch, pressure, and the chemical environment, including inflammatory mediators. The PSNs in the PG and NG connect centrally with the NTS neurons in the brainstem (Box 1), forming the afferent component of reflex circuits responsible for regulating vital functions such as heart rhythm and peristalsis. Recent research unveiled a significant and unanticipated variety of neurons within the NG [68]. There have been numerous subtypes identified, which are responsible for controlling the respiratory, gastrointestinal, and cardiovascular systems. These subtypes include mechanoreceptors that sense stretch and volume, baroreceptors, and receptors that respond to chemicals and nutrients [49,67]. BDNF immunoreactive neurons are detected in both PG and NG and are, respectively, about 50% (PG) or 80% (NG) of the total neurons in these ganglia [40]. BDNF immunoreactivity coexists with CGRP or TRPV1 immunostaining, but there are significant differences between the percentages of double-stained neurons in the two ganglia [40]. In the PG, BDNF-positive neurons are specifically identified as chemoafferent neurons [69].

#### 2.1.2. Second-Order Neurons

Second sensory neurons (SSNs) in somatic and visceral sensory pathways receive information from PSNs and relay it to other neurons along these pathways (Box 1). Whereas the SSN spinothalamic neurons do not express BDNF, somatic (in the spinal nucleus of the trigeminal nerve—SNTN) and visceral (in the nucleus tractus solitarius—NTS) SSN neurons express the BDNF protein and mRNA in the normal adult rat CNS [33]. In this study, the SNTN contains a few lightly stained scattered cells, whereas the NTS only rare moderately positive cells. Remarkably, in both locations, there is also a modest (SNTN) or heavy (NTS) presence of BDNF-immunoreactive fibers. In the NTS, BDNF is demonstrated to tonically regulate blood pressure, heart rate, and sympathetic nervous system activity in vivo. In the same study, it is also reported to negatively modulate excitatory synaptic transmission postsynaptically at glutamatergic primary afferent synapses [70].

#### 2.1.3. Higher-Order Neurons

The localization of BDNF in higher-order neurons along pain-related pathways is not fully characterized. In their seminal paper, Conner and colleagues published a semiquantitative detailed map of the distribution of the BDNF protein, its mRNA, and trkB in the CNS of normal adult rats [33]. The following description is mainly based on their data.

##### Brainstem

Within the brainstem, besides the aforementioned nuclei containing immunoreactive SSNs, the PBN is reported to contain no BDNF-immunoreactive neurons but to be particularly enriched in positive fibers [33]. In this location, there is a moderate number of cells expressing the BDNF mRNA. The lateral PBN sends a strong contingent of BDNF immunoreactive fibers to the AMYG central nucleus and has recently been demonstrated to be directly involved in the control of neuropathic pain behaviors [71]. This conclusion was reached after the observation that activation of glutamatergic or inhibition of GABAergic neurons in the lateral PBN by optogenetics induces neuropathic pain-like behavior in mice and that inhibition of glutamatergic neurons diminishes both basal nociception and neuropathic pain behaviors. In another study, the localized deletion of BDNF in the PBN shows that the basal thresholds of thermal and mechanical nociceptive responses are not altered, but animals with a deletion of the BDNF gene exhibit a reduction in the pain-relieving properties of morphine [72].

##### Amygdala

The complex of the amygdaloid nuclei displays a variegate pattern of BDNF immunostaining. Remarkably, the AMYG central nucleus is devoid of the BDNF protein and mRNA but contains a very dense network of heavily stained BDNF-immunoreactive fibers originating from the PBN [33]. The AMYG basolateral nucleus contains occasional or moderate numbers of positive cells. BDNF immunoreactivity was shown to increase in AMYG neurons after stress-induced visceral hyperalgesia [73].

##### Cerebral Cortex

BDNF-immunoreactive cells are localized in several cortical areas related to pain perception (Box 1). The insular and cingulate cortices contain low-to-moderate numbers of immunoreactive cells, as well as moderate levels of expression of the BDNF mRNA. There are no BDNF-immunoreactive fibers in cortical areas.

Expression of BDNF is upregulated in certain conditions of pathologic pain in cortical neurons. After CFA injection, BDNF levels in neurons (NeuN+) and glia (see below) are markedly increased in ACC and S1 [74]. In the same study, administration of recombinant BDNF by injections into the ACC, or injecting a viral vector engineered to synthesize BDNF into either the ACC or S1, results in increased neuronal excitability, as seen by increased LTP and persistent sensitivity to pain. Cyclotraxin-B, a powerful antagonist specifically targeting trkB signaling (see Section 4 Implications for Pain Management), effectively halts neuronal hyperexcitability, the development of cold hypersensitivity, and passive avoidance behavior in the ACC. These findings indicate that the development and persistence of the emotional component of chronic pain are strongly influenced by BDNF-dependent neuronal plasticity in this cortical area. In another study, the BDNF/trkB-mediated signaling pathway in the rostral ACC is shown to be involved in the development of neuropathic spontaneous pain-related aversion upon activation of NMDA NR2B receptors [75]. It has also very recently been reported that neonatal pain induced by intraplantar CFA injection or repetitive needle prick stimuli, in adulthood, results in impaired hearing associated with a decrease in BDNF immunoreactivity in neurons (and glia—see below) of the primary auditory cortex (Au1) in parallel with increased dendritic spine density, whilst no damage or synapse loss is found in the cochlea [76]. Oxycodone, a semi-synthetic opioid, attenuates hearing loss and alterations in Au1, and a trkB agonist reverses neonatal pain-induced hearing impairment and decreases caspase-3 expression. On the other hand, administration of a trkB antagonist in naïve mouse pups damages hearing development and increases caspase-3 expression. Therefore, these observations altogether confirm the modulatory action of BDNF in central auditory pathways.

Finally, a study in rats with peripheral inflammation demonstrates that continuous and spontaneous pain interferes with the connections between the ventral hippocampal CA1 region and the infralimbic cortex, as well as the ability of the hippocampus to regulate neuronal activity in the latter [77]. Targeted upregulation of BDNF in this circuit counteracts electrophysiological alterations, alleviates spontaneous pain, and expedites total recuperation from inflammatory pain.

**Table 1 biomolecules-14-00539-t001:** Distribution of BDNF in somatic pain pathway neurons after immunocytochemical staining.

Location	Type of Neurons	Function	Ref
DRGs	Small- to medium-size PSNs	Nociception	[54,55,56]
DRGs	Small- to medium-size PSNs	Inflammatory nociception	[78]
JG (X)	Small- to medium-size PSNs	Nociception	[40]
TG (V)	Small- to medium-size PSNs	Nociception	[64,65]
DRGs	Large-size PSNs	Nociception (after injury)	[59]
SNTN	SSNs	Localization study	[33]
CN	SSNs	Localization study	[33]
PBN	Projection neurons	Nociception	[33]
S1	Cortical neurons	Inflammatory/neuropathic pain	[74]
Au1	Cortical neurons	Chronic pain	[76]
ACC	Cortical neurons	Inflammatory/neuropathic pain	[74,75]

Abbreviations: ACC = anterior cingulate cortex; Au1 = primary auditory cortex; CN = cuneiform nucleus; DRGs = dorsal root ganglia; JG = jugular ganglion; PSNs = primary sensory neurons; S1 = primary somatosensory cortex; SNTN = spinal nucleus of the trigeminal nerve; SSNs = second-order sensory neurons. TG = trigeminal ganglion. Roman numerals indicate the cranial nerves.

**Table 2 biomolecules-14-00539-t002:** Distribution of BDNF in visceral pain pathway neurons after immunocytochemical staining.

Location	Type of Neurons	Function	Ref
DRGs	Small- to medium-size PSNs	Nociception	[79]
NG (X)	Small- to medium-size PSNs	Chemoafferent neurons	[40,69]
PG (IX)	Small- to medium-size PSNs	Chemoafferent neurons	[40,69]
NTS	Small- to medium-size SSNs	Modulation of cardiovascular afferents	[33,70]
AMY	Projection neurons to PBN	Hyperalgesia	[73]

Abbreviations: AMY = amygdala; DRGs = dorsal root ganglia; NG = nodose ganglion; NTS = nucleus tractus solitarius; PBN = parabrachial nucleus; PG = petrosal ganglion; PSNs = primary sensory neurons; SSNs = second order sensory neurons. Roman numerals indicate the cranial nerves.

### 2.2. Neuronal Mechanisms

The investigations on the neuronal mechanisms by which BDNF could regulate nociception and pain have concentrated on spinal and (to a lesser extent) trigeminal nociceptive pathways at the synapses between PSNs and SSNs. Many of these studies have been carried out in vivo, but ex vivo studies were also used to better explore the intervention of BDNF as a pain modulator. However, much less information is available regarding higher-order neurons along the pain transmission pathways described in Box 1. Broadly speaking, BDNF plays a pivotal role in synaptic plasticity within the spinal cord and the SNTN, where nociceptive signals are first processed. At PSN/SSN synapses (Figure 2), BDNF enhances glutamate release and modifies the function of AMPARs and NMDARs, resulting in amplified nociceptive signals [80,81] through a mechanism of LTP that is of pivotal importance in the genesis of central sensitization (see Section 3.2 Central Sensitization).

**Figure 2 biomolecules-14-00539-f002:**
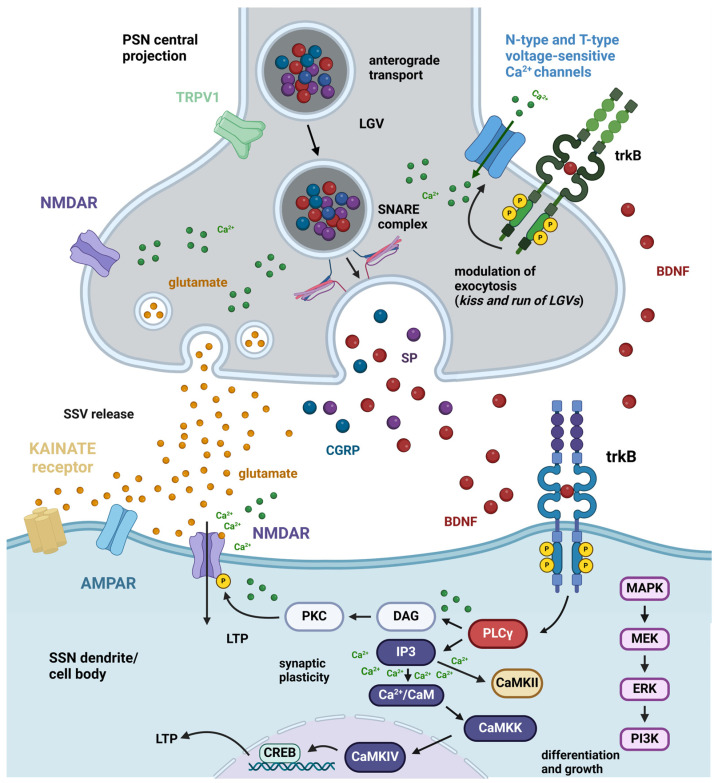
BDNF pre- and postsynaptic neuronal mechanism. The interface between the central projection of a PSN and an SSN dendrite/cell body is shown with special reference to the spinal cord dorsal horn. In C-type peptidergic primary afferent fibers originating from PSNs, there is a coexistence of glutamate as the main fast-acting neurotransmitter and some high molecular weight slow-acting neurotransmitters of peptide nature. The former is stored in SSV, whereas peptides colocalize in LGVs. Specifically, BDNF has been shown to colocalize with SP and CGRP, two of the most widely investigated sensory neuropeptides [23]. The two peptides also act as pain modulators and contribute to the excitation of the postsynaptic neurons and nearby neurons in the dorsal horn following volume transmission (see [53] for discussion). Along the central projections of the primary afferent fibers, BDNF is anterogradely transported to terminals. A subgroup of these terminals in the spinal cord dorsal horn expresses presynaptic (green) trkB receptors, but another subgroup does not express trkB [53]. For simplicity, an axo-dendritic contact is shown expressing trkB receptors at both pre- and postsynaptic sites. Under basal conditions and nociceptive pain, besides glutamate, C-type fibers originating from PSNs also release BDNF. The effects of BDNF in this case are due to the activation of postsynaptic trkB receptors (cobalt). There are three main pathways activated by these receptors within the cell: the MAPK/ERK cascade, which is mainly responsible for differentiation and growth but also intervenes in certain forms of pathological pain (see main text), the PLCγ/DAG/PKC pathway that phosphorylates postsynaptic NMDARs with an increase in NMDA Ca^2+^ currents, and the PLCγ/IP3 cascade that leads to a release of Ca^2+^ from internal stores and activation of calcium/calmodulin-stimulated protein kinases (CAMKs), eventually leading to synaptic plasticity and long-lasting LTP (see main text). When excitatory neurotransmission is reinforced, BDNF can also act on presynaptic trkB receptors, leading to an increase of Ca^2+^ in the presynaptic terminal across N-type and T-type voltage-sensitive Ca^2+^ channels [82], resulting in the augmented vesicular release of glutamate, as well exocytosis of LGVs. Abbreviations: AMPAR = α-amino-3-hydroxy-5-methyl-4-isoxazole propionic acid receptor; BDNF = brain-derived neurotrophic factor; Ca^2+^/CaM = calcium/calmodulin; CaMKII = calcium/calmodulin-stimulated protein kinase II; CaMKIV = calcium/calmodulin-stimulated protein kinase IV; CGRP = calcitonin gene-related peptide; CREB = cAMP response element-binding protein; DAG = diacylglycerol; ERK = extracellular signal-regulated kinase; IP3 = inositol 1,4,5-trisphosphate; LGV = large granular vesicles; LTP = long-term potentiation; MAPK = mitogen-activated protein kinase; MEK = mitogen-activated protein kinase kinase; NMDAR = N-methyl-D-aspartate receptor; PI3K = phosphatidylinositol 3-kinase; PKC = protein kinase C; PLCγ = phospholipase C γ; PSN = primary sensory neuron; SNARE = soluble N-ethylmaleimide-sensitive-factor attachment protein receptor; SP = substance P; SSN = secondary sensory neuron; trkB = tropomyosin receptor kinase B; TRPV1 = transient receptor potential vanilloid. Created with BioRender.com.

Remarkably, BDNF exerts a neuromodulatory effect under both uninjured (normal) and injured conditions [83]. Multiple lines of evidence suggest that BDNF primarily functions as a pro-nociceptive modulator, playing a key role in the onset and persistence of inflammatory, chronic, and/or neuropathic pain [29]. However, there have been sporadic reports of BDNF having anti-nociceptive effects, indicating that its actions can vary depending on specific cell types, molecular pathways, and pathological circumstances [25,83,84,85,86]. Nonetheless, it was also reported that BDNF produced by primary afferents has minimal relevance in the processing of pain and itch [87]. All these issues will be discussed in the following sections.

#### 2.2.1. Ex Vivo Studies on Nociceptive/Pain Modulation by BDNF

Ex vivo studies allowed the effects of endogenous or pharmacologically administered BDNF to be dissected on acute slices or slice organotypic cultures [88]. Most of these studies were carried out on spinal cord slices and were focused on the substantia gelatinosa (lamina II of the gray matter). It is worth mentioning here that neurons in the two types of preparations display similar electrophysiological properties [89]. It is also useful to recall that the ex vivo approach allows for the performance of electrophysiological recordings and other functional studies (e.g., Ca^2+^ imaging) that have been useful in ameliorating our understanding of the circuitry involved in BDNF modulatory functions. I will in this section describe only the results related to slices/organotypic cultures prepared from normal untreated animals. When these preparations were obtained from animals previously treated in vivo, they will be discussed in the following sections.

Acute slices of the rat spinal substantia gelatinosa have been used to dissect the role of BDNF as a pain modulator, focusing on the synapses between PSNs and SSNs [56]. In this study, two series of experiments were performed. In the first, BDNF was applied to naïve slices at pharmacological concentrations; in the second, inflammation was mimicked after capsaicin administration to slices. After patch-clamp recordings, calcium imaging, and immunocytochemistry on slices from 8-12 days post-natal rats, it was demonstrated that BDNF enhances the release of sensory neurotransmitters (glutamate, substance P, CGRP) in lamina II by acting on presynaptic trkB receptors expressed by TRPV1 immunoreactive peptidergic primary afferent fibers. Ib1-immunoreactive microglial cells did not stain for BDNF in this study. However, the BDNF released by microglia has been implicated in the generation of neuropathic pain in other in vivo reports (see Section 2.3.1 Glial Cells—Microglia). BDNF also enhances the excitatory responses to NMDA in the rat spinal cord [35]. The underlying mechanism may include enhancing the activity of presynaptic NMDARs on primary afferent terminals, leading to an increase in the transmission of excitatory glutamatergic signals [90].

It was also shown that exposing substantia gelatinosa neurons to BDNF in organotypic culture for a period of 5 to 6 days results in an electrical pattern that closely resembles the one induced by CCI in live subjects [89,91]. Both BDNF and CCI enhance the amplitude and frequency of spontaneous excitatory postsynaptic currents (sEPSC) and miniature excitatory postsynaptic currents (mEPSC) in neurons that exhibit delayed firing. Furthermore, a more comprehensive examination reveals that both interventions also disclose the existence of a previously unidentified group of large mEPSC events, which are not present in neurons from rats after sham surgery or in control organotypic cultures [92,93].

As will be discussed in the following sections, peripheral nerve damage stimulates the secretion of BDNF from primary afferent terminals and spinal microglia, leading to increased excitability of the dorsal horn, which plays a role in central sensitization and the initiation of neuropathic pain (see Section 3.2 Central Sensitization). While it is widely acknowledged that the reduction of inhibition driven by γ-amino-butyric acid (GABA) or glycine is responsible for the onset of neuropathic pain [94], some research indicates that BDNF administration increases the K^+^-induced release of GABA in the dorsal horn [95]. BDNF also increases spontaneous GABA release (as shown by increases in miniature inhibitory postsynaptic current (mIPSC) frequency) yet reduces stimulation-evoked GABA release (i.e., reduces the amplitude of evoked IPSCs), very likely regulating the release of the inhibitory transmitter from the islet cells [96], a long-known population of inhibitory neurons in lamina II. To shed more light on these conflicting results, rat spinal cord neurons were challenged with BDNF for six days in a defined-medium organotypic culture to replicate the alterations in spinal BDNF levels that occur with peripheral nerve injury [97]. Very interestingly, it was demonstrated that BDNF enhances the amplitude of GABAergic and glycinergic mIPSCs in both inhibitory tonic islet cells and tonic central neurons and excitatory delay radial neurons (for classification and properties of substantia gelatinosa neurons, see [53]). Furthermore, the neurotrophin enhances the amplitude and frequency of sIPSCs in presumed excitatory neurons, whereas it decreases the amplitude of sIPSCs in inhibitory neurons, although the frequency of these events remains unchanged. From these results, it was concluded that the dual effect of BDNF on GABAergic transmission (the increase in inhibitory input to excitatory neurons and the decrease in inhibitory input to inhibitory neurons) only apparently contradicts the general notion that BDNF enhances the overall excitability of the dorsal horn because the effect of BDNF in the substantia gelatinosa may be primarily influenced by the enhancements in excitatory synaptic transmission, rather than by the inhibition of inhibitory transmission [97]. Along this line of thinking, some time ago, I proposed a neuronal circuitry to explain the apparent dual effect of BDNF in lamina II whereby inhibition of inhibitory transmission occurs under conditions of long-lasting pain and central sensitization [53].

In a 2022 paper, both rat and human acute spinal cord slices were treated with recombinant BDNF to investigate the possible occurrence of sexual dimorphism in neuronal hyperexcitability under pathological pain conditions [98]. The administration of ex vivo BDNF did not have any impact on synaptic NMDAR responses in female lamina I neurons of rats. Crucially, this disparity in the way spinal pain is processed is maintained across species, from rats to humans. Like rats, the application of BDNF to spinal cord slices obtained from human donors results in a decrease in neuronal markers of reduced inhibition and an increase in markers of enhanced excitation in cells of the superficial dorsal horn. This effect is observed in males but not females. Notably, ovariectomy in female rats replicates the neuronal phenotype of pathological pain observed in males.

#### 2.2.2. In Vivo Studies on Inflammatory Pain Modulation by BDNF and proBDNF

##### BDNF

Numerous in vivo studies have concluded that BDNF is an important pro-nociceptive modulator in several forms of inflammatory pain. The release of BDNF by sensory neurons and their primary afferent axons was demonstrated to be essential for the development of inflammatory pain caused by both formalin and carrageenan [99]. The pro-nociceptive impact of BDNF on inflammatory pain is commonly linked to pre- and post-synaptic enhancement of glutamatergic transmission in the spinal dorsal horn through NMDAR plasticity. BDNF enhances nociceptive spinal reflex activity and triggers c-fos expression in dorsal horn neurons in an NMDAR-dependent manner when rats are treated with carrageenan [35]. In the same paper, it was discovered that the molecule trkB-IgG, which sequesters BDNF, is capable of alleviating hyperalgesic behavior and restoring normal function. In mice, the combination of BDNF and the NMDAR antagonist D-2-amino-5-phosphonovaleric acid (D-APV) stops the development of BDNF-induced hyperalgesia [100]. The enhancement of NMDA neurotransmission is facilitated by the activation of protein kinase C (PKC)- and phospholipase C (PLC)-dependent pathways [80,101].

It was long ago observed that BDNF can cause hyperalgesia by acting presynaptically at the synapses between PSNs and SSNs in the spinal dorsal horn. In a rat model of inflammation with CFA subcutaneous injections, it was demonstrated that BDNF enhances the presynaptic release of glutamate in lamina II neurons [102]. The observed impact is linked to a stronger synaptic input from large myelinated Aβ afferent fibers. These fibers are known to normally encode light touch and may therefore be responsible for the development of allodynia in this model of inflammatory pain. In keeping with these observations, CFA- and TNFα-induced peripheral inflammation was observed to increase the in vivo expression of BDNF, trkB, and TRPV1 in DRGs [60]. In the same study, DRG cultures that are treated with TNFα over a long time exhibit a notable increase in the amounts of mRNA and protein for BDNF and trkB. Additionally, there is an increase in the release of BDNF and the activation of the phospho-extracellular signal-regulated kinases 1 and 2 (ERK1/2) signal pathway induced by trkB. Furthermore, the production of CGRP and substance P increases following prolonged exposure to TNFα or immediate activation with BDNF [60]. Altogether, these observations are fully confirmatory of our data in spinal cord slices subjected to capsaicin challenge to mimic inflammation ex vivo (see above).

There may be several pathways driving neuronal BDNF signaling in response to inflammatory conditions. In one study, ATP-gated purinergic receptor 4 (P2RX4) expression increased in nociceptive neurons during prolonged peripheral inflammation and colocalized with BDNF in these neurons [103]. P2RX4-deficient mice exhibit defective BDNF-dependent signaling pathways, including phosphorylation of ERK 1/2 and the GluN1 subunit of the NMDAR, as well as downregulation of the K-Cl co-transporter 2 (KCC2) in the dorsal horn of the spinal cord. It is worth noting that to ascertain if sensory neurons were the primary cells responsible for producing BDNF in the dorsal horn during prolonged inflammation, the authors examined if the paw injection of CFA induces microglial activation and if BDNF is expressed by spinal microglia to conclude that no apparent alterations in the structure or quantity of ionized calcium-binding adaptor molecule 1 (Iba1)^+^ microglia are found in the dorsal horn when comparing the control and CFA conditions. These findings indicate that P2RX4, which is expressed by PSNs, regulates the release of neuronal BDNF, hence contributing to increased neuronal excitability in the context of chronic inflammatory pain [103].

Recently, the role of peripheral BDNF in inflammatory pain has been reevaluated using a conditional Advilin-CreERT2 knock-out mice model. This mouse selectively lacks BDNF in sensory neurons [29,87]. Both investigations using this model noted a decrease in the nocifensive response during the second phase of the formalin test, indicating a decrease in central sensitization. However, this effect is only detected in males [87]. Interestingly, these authors did not observe any disparity between sexes in the progression of pain hypersensitivity in the CFA model of chronic inflammatory pain [87]. However, another group discovered that the deletion of BDNF has a significant impact on hyperalgesic priming [29]. This is a neuroplastic mechanism that creates a hidden state of reinforced sensitivity in sensory neurons, which is believed to facilitate the shift from acute to chronic pain (see Section 3 BDNF and Neuronal Sensitization). Induction of hyperalgesic priming was achieved with the intraplantar injection of carrageenan, and the subsequent effect was revealed by the injection of prostaglandin E2 (PGE2) after six days. While control mice that are prepared in advance demonstrate long-lasting reinforced sensitivity to pain following the injection of PGE2, animals devoid of BDNF in their sensory neurons only show a temporary increase in their ability to sense pain [29].

Another recent study has confirmed the intervention of BDNF in inflammatory (and neuropathic—see below) pain after assessment of the effects of 1-(1, 1-dimethyl ethyl)-3-(1-naphthalenyl methyl)-1H-pyrazole [3,4-d] pyrimidine-4-amine (1NMP), a cell-permeable kinase inhibitor blocking trkB autophosphorylation and signaling [104]. Inflammatory pain induced by formalin in transgenic TrkBF616 mice expressing a mutated trkB that displays sustained signaling in the absence of the inhibitor is strongly reduced after 1-NMP treatment.

Similar mechanisms to those described in the DRGs and spinal cord have been identified in the TG neurons and SNTN receiving the peripheral stimuli from the orofacial territories and the dura mater (see Box 1). The intervention of BDNF (and other neurotrophins) in modulating the trigeminal pain pathways has been very recently reviewed, and the authors highlighted that research on the head sensory territories has been minor, likely due to a relative lack of animal models in comparison to those available for spinal pain [105]. Yet one must keep in mind that the laminar structure of the SNTN is very similar to that of the spinal dorsal horn, considering that both derive from the alar plate of the embryonic neural tube developing into the dorsal gray column. The alar plate of the brainstem is located lateral to the sulcus limitans in the floor of the rhomboid fossa and gives rise to the sensory-receiving nuclei of the adult brainstem, among which are the sensory nuclei of the trigeminal nerve. Thus it is not surprising that most mechanisms of BDNF modulation of trigeminal nociceptive pathways are similar to those described for the spinal cord dorsal horn, particularly the substantia gelatinosa [53].

The SNTN is physically organized into three subdivisions: pars oralis, interporalis, and caudalis [106]. The pars oralis refers to the endpoint of incoming connections from the mouth and nose areas. It includes circuits that play a role in brain stem reflexes. The pars interpolaris gets sensory information from the same side of the face and projects with the pars oralis to create the ascending anterior trigeminothalamic tract. The pars caudalis also receives sensory input from the face, forehead, cheek, and jaw and is responsible for transmitting temperature sensation from these regions.

Hyperalgesia associated with BDNF/trkB signaling in TG neurons projecting to the cranial/interpolaris transition zone of the SNTN has been studied after masseter muscle inflammation [107]. BDNF/trkB immunoreactive small/medium-diameter TG neurons are noticeably more abundant in inflamed animals compared to naïve rats, and most dissociated small-diameter TG neurons display a depolarization response to BDNF that was coupled with spike discharge in whole-cell current-clamp registrations. As expected, BDNF-induced TG neuron excitability alterations are eliminated by the pan-trk inhibitor K252a. It is worth noting that in this study, the neurons responsible for trigeminal inflammatory hyperalgesia express both BDNF and trkB, suggesting the possibility of an autocrine effect of the neurotrophin, as described earlier, after ex vivo studies on the spinal substantia gelatinosa (see Section 2.2.1. Modulation of Inflammatory Pain by BDNF—Ex vivo Studies). Expression of trkB in mouse lingual sensory afferents (i.e., the peripheral branches of the TG neurons giving rise to the lingual nerves) was demonstrated in a paper where oral squamous carcinoma cells were shown to release BDNF and contribute to oral cancer pain via peripheral trkB activation [108]. The cellular mechanism by which BDNF causes TG neuron hyperexcitability was examined in culture-isolated neurons. Exogenously administered BDNF was shown to increase T-type Ca^2+^ currents by stimulating the trkB-phosphoinositide 3-kinase (PI3K)-p38-protein kinase A (PKA) intracellular pathway [109]. Furthermore, an upregulation of the *Bdnf* gene in mouse TG neurons is observed in a model of mechanical irritation and localized inflammation in the periodontium, mimicking the pain derived from orthodontic forces in most patients [110]. Remarkably, intraganglionic injection of resiniferatoxin (a potent functional analog of capsaicin) before the application of an orthodontic force reduces mice’s orthodontic-induced nocifensive behavior, indicating that, also in TG neurons, TRPV1 receptor expression is pivotal for the genesis of inflammatory pain as observed in DRG neurons [111].

Some other studies have analyzed the expression of BDNF in the SNTN neurons following different experimental forms of inflammatory pain. After subcutaneous capsaicin injection in the upper lip, SSNs in SNTN display a reduction in BDNF immunoreactivity [112]. In this study, orexin-A lowers capsaicin-induced trigeminal pain through the modulation of pain effects on BDNF and cyclooxygenase-2 (COX2) expression, but how TG neurons activation by capsaicin influences the expression of BDNF in SNTN neurons was not investigated. In another paper, inflammatory pain was generated by injecting CFA into the rat temporomandibular joint to study the effects of transcranial direct-current stimulation (tDCS) as a therapeutic tool [113]. Joint inflammation raises the level of brainstem BDNF after ELISA immunodetection, and tDCS reverses such an effect. A third study evaluated the effects of PACAP6-38, a pituitary adenylate cyclase-activating peptide (PACAP) type I receptor (PAC1R) antagonist, in nitroglycerin-induced central sensitization, after stereotaxic injection of the organic nitrate into the SNTN, a chronic migraine model [114]. It was thus demonstrated that the effects of nitroglycerine can be reversed by inhibition of PAC1R and downregulation of the CREB/BDNF signaling pathway.

Whereas the studies above converged to indicate the primary role of neuronal BDNF in the genesis of inflammatory trigeminal pain, in a trigeminal allodynia rat model, which closely mimics chronic migraine, P2X4 receptors, and associated signaling pathways were examined to conclude that it is the SNTN microglial cells expressing BDNF to upregulate these receptors [115]. Blocking the purinergic P2X4 receptor inhibits the expression of BDNF and other molecules such as p38, excitatory amino acid transporter 3 (EAAT3), c-Fos, and CGRP, causing anti-nociception.

##### proBDNF

Although the intervention of BDNF in inflammatory pain is widely accepted, the role of proBDNF, which is also expressed in nociceptive pathways, has been comparatively little investigated.

In rodents, proBDNF acts as an inflammatory mediator in pain regulation [116]. While it is well established that BDNF is produced by PSNs and thereafter released in an activity-dependent manner, the mechanisms of proBNF secretion remain to be understood in full [117]. It was nonetheless shown in vitro that the activity-dependent release of BDNF could be regulated by the interaction of sortilin with proBDNF [118]. In vivo, the expression of BDNF and sortilin is dramatically elevated in the ipsilateral L4/5 DRGs following the injection of CFA into the rat hind paw, and the two molecules are found to be colocalized [119]. Knocking down sortilin 1 in the L5 DRG using in vivo an adeno-associated virus effectively reduces the pain-like response. Notably, a synthetic peptide, corresponding to the sequence of amino acid residues 89–98 of proBDNF, was discovered to disrupt the interaction between proBDNF and sortilin when compared to a randomly arranged peptide. This synthetic peptide also hinders the release of BDNF in vitro in response to activity and decreases the occurrence of CFA-induced mechanical allodynia and heat hyperalgesia in vivo. The synthesized peptide also disrupts the capsaicin-induced activation of ERK in the spinal cord of rats injected with CFA [119].

Recently, the importance of the proBDNF/p75^NTR^ signaling pathway in inflammatory and rheumatoid arthritis pain was studied in mice and human patients to conclude that proBDNF/p75^NTR^ boosts rheumatoid arthritis and inflammation by triggering proinflammatory cytokines [120]. In this study, a higher degree of synovitis in the synovial membrane of patients with rheumatoid arthritis compared to the control group with osteoarthritis was observed. The mRNA levels of p75^NTR^ and sortilin, as well as the protein level of proBDNF, are considerably higher in the peripheral blood mononuclear cells of rheumatoid arthritis patients compared to healthy donors. ELISA consistently demonstrates elevated levels of p75^NTR^, sortilin, TNFα, interleukin-1β (IL-1β), interleukin-6 (IL-6), and interleukin-10 (IL-10) in the serum of patients with rheumatoid arthritis compared to healthy subjects. Expression of the proBDNF/p75^NTR^ signaling pathway molecules and elevated levels of inflammatory cytokines are also observed in the spinal cord by utilizing a widely accepted collagen-induced arthritis mouse model. It was reported that after CFA-induced inflammation in male rats, there is a rapid increase of proBDNF in NeuN-positive neurons and GFAP-positive astrocytes in the spinal cord ipsilateral to the CFA injection site [121]. In addition, the intrathecal administration of a cleavage-resistant proBDNF protein decreases thermal and mechanical pain thresholds, whereas a monoclonal anti-proBDNF antibody reduces CFA-triggered pain and inflammation. In the same study, the administration of CFA caused the activation of p75^NTR^ and its downstream signaling pathways, ERK1/2 and nuclear factor-kappa B (NF- κB) p65, in the spinal cord, and the intrathecal administration of p75^NTR^ extracellular domain effectively inhibited pain and neuroinflammation generated by CFA, leading to the conclusion that up-regulation of proBDNF/p75^NTR^ signaling also contributes to inflammatory pain generation in the spinal cord [121].

#### 2.2.3. In Vivo Studies on Neuropathic Pain Modulation by BDNF

DRG/Spinal cord

BDNF is also responsible for the maladaptive changes in neural circuits that occur in neuropathic pain. For this type of pain, most observations again refer to the synapses between PSNs and SSNs in the dorsal horn of the spinal cord. Research conducted long ago on rats and mice with a partial sciatic nerve injury revealed that spinal BDNF plays a role in the changes that occur in pain pathways because of nerve damage [122,123,124,125]. These studies showed that the administration of either anti-BDNF or anti-trkB neutralizing antibodies, trk inhibitors, or the BDNF scavenger trkB/Fc through the bloodstream eliminates the behavioral responses to the increased pain sensitivity. Several other reports documented that increased expression of BDNF in PSNs and its subsequent release at their central processes in the spinal cord dorsal horn occurs in several models of nerve damage, such as sciatic nerve transection, CCI, and nerve ligation [126,127,128,129,130,131,132]. More recently, expression changes in TRPV1 receptors, CGRP, and BDNF were observed in rat DRGs after resiniferatoxin (a very strong agonist of TRPV1 receptors)-induced neuropathic pain, with a reduction in CGRP and TRPV1 and increase in BDNF leading to the development of pathologic pain [111].

Under neuropathic pain conditions, BDNF induces hyperexcitability in spinal neurons by enhancing the action of NMDARs, as observed in inflammatory pain. When rats with spinal nerve ligation are given BDNF through an injection into the spinal cord, it causes LTP and increased sensitivity to touch by activating the NMDA-NR2BRs [126,133]. Simultaneous changes in inhibitory and excitatory transmission interact and increase the sensitivity of central sensory neurons, leading to the so-called central sensitization of these neurons—see Section 3.2 Central sensitization. Specifically, it was documented that the decrease in KCC2 caused by BDNF is both enough and essential to facilitate NMDA-NR2BR phosphorylation and the initiation of LTP in lamina I neurons of rats with nerve injuries [134]. The STEP61 phosphatase plays a crucial role in the communication between BDNF-dependent Cl^−^ dysregulation and NMDAR phosphorylation. When STEP61 phosphatase is downregulated, it leads to a decrease in KCC2 levels and ultimately results in the enhancement of neuronal excitation [135]—see also Section 2.2.1 Modulation of Inflammatory Pain by BDNF—Ex vivo Studies.

While the mechanisms mentioned above primarily rely on postsynaptic changes at PSN-SSN synapses, there have also been reports of BDNF-mediated presynaptic effects. In rats with CCI, the expression of the Cl^−^ transporter NKCC1 in DRGs is enhanced by BDNF, leading to increased intracellular Cl^−^ levels with temporary GABAergic presynaptic inhibition and enhanced nociception [136]. The presynaptic effects are probably influenced by microglial BDNF, which has been demonstrated to enhance the NMDA-mediated responses by acting on afferent terminals [90]—see also Section 2.3.1 Glial Cells—Microglia.

A recent study has examined the effects of 1NMP treatment in TrkBF616 mice (see also Section 2.2.1 Modulation of Inflammatory Pain by BDNF—In vivo Studies DRG/Spinal Cord) in an experimental model of neuropathic pain [104]. Mice were subjected to spinal cord injury (SCI), and weekly evaluation of tactile sensitivity using the von Frey test revealed that administering 1NMP immediately after SCI prevents the onset of mechanical hypersensitivity for a duration of up to 4 weeks. In contrast, when therapy commences 2 weeks after injury, 1NMP only partially reduces the hypersensitivity of the hind paw. These findings indicate that the activation of trkB signaling soon after SCI plays a role in the development of maladaptive changes in neural connections, resulting in pain hypersensitivity [104]. Another proof of the importance of BDNF in neuropathic pain came from a novel study of the microRNA-489-3p regulation of the oncoprotein DEK [137]. DEK, a DNA-modifying nuclear protein, is involved in maintaining the overall structure and function of heterochromatin, as well as regulating processes such as transcription, DNA replication, and repair. DEK is widely expressed in CNS and, among others, has been proposed to have a role in neuropathic allodynia. SNL, another widely used model of neuropathic pain, was reported to induce a decrease in miR-489-3p expression, while increasing the expression of DEK, which recruits ten-eleven translocation methylcytosine dioxygenase (TET1) to the promoter fragments of the Bdnf gene, thereby enhancing its transcription in the dorsal horn [137]. SNL-induced neuropathic pain was also shown to be alleviated by the type-B monoamine oxidase (MAO-B) inhibitor, KDS2010, through competitively blocking the BDNF/trkB/NMDA-NR2B pathway [138], giving further support to the intervention of this pathway in the genesis of neuropathic pain. This concept is further validated in another investigation, which demonstrates a substantial increase in BDNF and trkB immunoreactivity in the spinal dorsal horn after a burn injury that was responsible for the development of both inflammatory and neuropathic pain [139]. This study found that the intrathecal administration of trkB-Fc, which is a scavenger of endogenous BDNF, decreases mechanical allodynia. Additionally, the administration of resolvin D1 (RvD1), which is a natural lipid modulator produced during the end of acute inflammation and has powerful anti-inflammatory and pro-resolution effects, prevents the increase in BDNF and trkB associated with injury.

##### Trigeminal System

A mouse model of orofacial neuropathic pain, the inferior alveolar nerve (IAN) transection that causes whisker pad ectopic allodynia can be used to assess trigeminal pain. IAN transection increases the peripheral synthesis of BDNF. However, the regulation of NMDARs in peripheral and central sensitization mechanisms that follow the procedure shows different mechanisms than those described above that were centered upon the activation of the NMDA-NR2BRs (see Section 2.2.3 Modulation of Neuropathic Pain by BDNF). Notably, neither high- nor low-dose NMDA increases TG BDNF levels after IAN transection [140]. In TG neurons, NR2AR deletion significantly blocks IAN transection-induced alterations in BDNF and other molecules. However, NR2BR knockout does not affect BDNF. Thus, NR2ARs and NR2BRs have opposing effects on BDNF and chemokine (C-C motif) ligand 2 (CCL2) production [140]. In CCI of the infraorbital nerve, another model of neuropathic pain that generates facial mechanical hyperalgesia, BDNF is upregulated in TG neurons, but observations at higher level neurons in the brainstem and cerebral cortex give puzzling results, as BDNF levels also increase in sham-operated animals [141]. Other authors used the CCI of the infraorbital nerve to model neuropathic pain in rats. They employed different doses of naltrexone to reverse facial mechanical allodynia but observed that the opioid antagonist is unable to modulate the levels of BDNF in the brainstem [142].

At the supraspinal level, it is notable that the CREB/BDNF pathway in the ACC regulates neuropathic pain and anxiety/depression-like behaviors in rats, as demonstrated after the reversal of pain hypersensitivity by central specific knockdown and peripheral inhibition of CREB [143].

As a final comment on the intervention of BDNF in spinal and trigeminal neuropathic pain, it is important to be well aware that while the role of BDNF is well-explained in mechanistic investigations such as those reported above, findings from human studies display inconsistency, and further research is required to assess the methodological difficulties associated with utilizing serum BDNF as a biomarker in neuropathic pain [144]—see Section 5 Clinical Trials.

### 2.3. BDNF in Glia and Other Cells of Relevance in Nociceptive/Pain Pathways—Localization and Mechanisms

Undoubtedly, the domain of pain research first focused on neurons. However, the current emphasis has also turned towards investigating the contribution of non-neuronal cells, specifically their interaction with neurons, which arises because of the response of these cells to a modified neuronal activity [145,146]. As I will discuss in the following, several types of glia and immune cells can synthesize BDNF (Figure 3) and thereby influence the activity of neurons in nociceptive/pain pathways.

#### 2.3.1. Glial Cells

Schwann cells [147] in the PNS, microglia [148,149,150], astrocytes [151,152], and oligodendrocytes [153] in the CNS synthesize and release BDNF. Broadly speaking, glial cells contribute to BDNF production during certain developmental processes and in response to inflammatory signals and injury in the mature nervous system [154,155]. The BDNF release from the glia can influence synaptic plasticity and pain sensitization [156].

The localization of glial cells expressing the BDNF protein in pain pathways is summarized in Table 3.

Schwann Cells

Schwann are the primary glial cells found in the PNS. Myelinating Schwann cells produce the myelin sheath, which provides electrical insulation to axons and enables the rapid conduction of action potentials by a process called saltatory conduction. Non-myelinating Schwann cells encircle unmyelinated fibers to create Remak bundles, in which Schwann cells wrap around several small-diameter axons, separating them from each other [163]. In addition to these structural tasks, Schwann cells interact with neurons, particularly with whole axons, to regulate their physiological functions. Schwan cells produce and release several mediators including BDNF, chiefly in response to nerve injury [164]. A recent in vitro study [157] has demonstrated that with the activation of a signaling cascade by which allopregnanolone synthesized by Schwann cells upregulates the synthesis and release of BDNF in an autocrine fashion, the latter eventually regulates protein kinase type Cε (PKCε) in DRG neurons, via trkB activation in a paracrine manner (Figure 3A). This finally leads to the sensitization of primary afferent nociceptors [165]. BDNF secretion by Schwann cells has also been postulated to occur in trigeminal neuralgia with a mechanism involving P2RX4 [166] in a similar fashion to that described in microglia (see below).

Microglia

Microglia are the resident CNS macrophages. However, the exact contribution of these cells to the immune defense of the CNS is uncertain due to the immunological privilege of the brain and spinal cord [167]. Research on microglia and pain is very extensive and has mainly focused on the spinal cord somatosensory pathways [168]. The intervention of microglial BDNF in neuropathic pain has been very recently authoritatively reviewed [169].

The first demonstration that spinal microglia produce BDNF to induce neuropathic pain dates back to 2005 [148]. It is accepted that microglia regulates neuropathic pain after peripheral nerve injury primarily as a consequence of C-fiber activation [170], but also with the contribution of A-fibers [171]. There are different classes of microglial activators involved in the regulation of spinal nociception [168]. A series of activators are at first released by PSNs. As a result, many microglial signal molecules are upregulated and contribute to supporting neuropathic pain. Mediators released by microglia then control synaptic plasticity and central sensitization. Specifically, among these mediators, BDNF is implicated in the disinhibition of spinal cord circuits and the development of neuropathic pain [146,148]. BDNF is secreted by microglia in response to nerve injury that leads to ATP release and activation of microglial P2RX4 (Figure 3B). Microglial BDNF then binds to trkB in lamina I neurons, resulting in the upregulation of intracellular Cl^–^ levels through the downregulation of neuronal KCC2 in male rats [172]. The accumulation of Cl^–^ in lamina I neurons reduces the effectiveness of GABA/glycine ionotropic transmission, leading to greater excitability and heightened sensitivity to pain because of the enhancement of synaptic GluN2B-NMDAR currents through the activation of Fyn kinase. This eventually primes the firing of action potentials, ultimately promoting the development of neuropathic pain [148]. Microglial-derived BDNF increases the excitatory drive to excitatory dorsal horn neurons with an intensification of spontaneous action potential discharge and inhibits that to inhibitory neurons by both presynaptic and postsynaptic mechanisms [173,174,175]. The absence of GABAergic inhibition allows non-nociceptive Aβ fiber-mediated excitatory signals to reach the outer layers of the spinal dorsal horn, this mechanism promoting the development of allodynia [176].

Very recently, the activation of microglia through the BDNF-trkB signaling pathway was also demonstrated to downregulate neuronal KCC2 in dynamic allodynia (caused by mild stimulation of the skin and mediated by low-threshold mechanoreceptors) that follows a sparing nerve injury [177]. In keeping with these observations, it was reported that minocycline, a partially synthetic derivative of tetracycline that inhibits the activation and replication of microglia without directly affecting astrocytes or neurons, increases the expression of KCC2 and GABA_A_/γ2 receptors but decreases BDNF expression in the lumbar spinal cord after nerve injury [178]. In the trigeminal system, neuropathic pain can be provoked by delivering ATP-stimulated microglia or BDNF intrathecally [140,141]. On the other hand, reducing BDNF in microglia, blocking trkB, or deleting P2RX4 expression [113,114,141] can oppose the onset of pain.

Yet, the identification of BDNF in spinal (and trigeminal) microglia poses a challenge, despite the straightforward detection of BDNF in DRG and TG neurons and primary afferents within the spinal cord and SNTN [53]—see Section 2.1.1. Primary Sensory Neurons. Nonetheless, it was recently demonstrated that induction of spinal LTP in the absence of nerve injury triggers long-lasting chronic pain with an increase in CGRP immunoreactive terminals in the dorsal horn and that behavioral and structural changes are prevented by blocking NMDARs, ablating spinal microglia, or conditionally deleting microglial BDNF [179]. In keeping with the observations converging to show the intervention of spinal microglia in the onset of neuropathic pain, a study conducted by Zhao et al. in 2006 found that BDNF, produced from nociceptors, plays a role in regulating acute and inflammatory pain but not neuropathic pain [99]. It was also demonstrated that sensory neuron-derived BDNF is an important factor in the progression from acute to chronic neuropathic pain [29]. These issues are also discussed in Section 2.2.1 Modulation of Inflammatory Pain by BDNF.

Emerging research has additionally indicated the existence of sexual dimorphism in the effects of microglia on nociception [180]. Although there is comparable proliferation of microglia in the dorsal horn of both males and females, it is noteworthy that females do not exhibit upregulation of P2RX4 under neuropathic pain conditions. Instead, they employ an alternative mechanism that is independent of microglia to mediate pain hypersensitivity. In females, changes in sensory processing in the dorsal horn involve the invasion of macrophages and T-lymphocytes [180]. Yet. as in males, this leads to attenuation of inhibition following the collapse of the Cl^−^ gradient. In females, the downfall of the Cl^−^ gradient is also provoked by CGRP [181] released from primary afferent terminals [182].

The intervention of microglial BDNF has been postulated not only in neuropathic pain but also in other forms of pain, such as, e.g., inflammatory and nociplastic pain. Two polarized forms of activated microglia, i.e., the proinflammatory M1 classically activated microglia and the anti-inflammatory M2 alternatively activated microglia are currently described (reviewed in [154]). BDNF was demonstrated to be released by M1 microglia, leading to enhanced neuroinflammation in nociplastic pain [154]. BDNF was also detected immunocytochemically in microglia (and astrocytes) in a model of inflammatory cystitis [183], in autoimmune prostatitis [159], during the early stage of diabetic neuropathy [158], and in a mouse model of chronic post-ischemic pain [184].

Release of BDNF from microglia is promoted, among others, by colony-stimulating factor 1 (CSF-1) [150,184,185] and/or Wnt3a [186]. Notably, CSF-1 mimics the effects of neuropathic pain in vivo, and its antagonist blockade or gene knockout alleviates neuropathic pain [187], and Wnt3a is downregulated after exercise when neuropathic pain is induced by sciatic nerve damage [188].

Astrocytes

Astrocytes are five times as numerous as neurons. They perform numerous fundamental activities in a healthy CNS. Reactive astrogliosis, a pathological characteristic of CNS structural lesions, occurs when astrocytes respond to CNS stressors. Astrocytes have been categorized into protoplasmic or fibrous subgroups based on their morphologies and anatomical placements [151]. Regulated increases in intracellular Ca^2+^ concentration [Ca^2+^]_i_ indicate astrocyte excitability [189]. A vast body of evidence shows that these controlled increases in astrocyte [Ca^2+^]_i_ are important for astrocyte–astrocyte and astrocyte-neuron communication [190]. Studies in vitro have shown that astrocytes produce BDNF under normal conditions [191] and after injury [192,193] (Figure 3C). Other in vitro studies have shown that the release of the neurotrophin is mediated by trkB.T1 [194], a truncated isoform of the trkB receptor that lacks the intracellular kinase domain of the full-length receptor and is upregulated in numerous CNS-damaged scenarios [195]. In addition, in vitro evidence was obtained that the inhibition of the inwardly rectifying potassium channel 4.1, an astrocytic channel that regulates neuronal activity, induces BDNF expression in these glial cells [196].

Pathological reactive astrocytes cause inflammation and neuropathies and, together with microglia and neurons, interact in spinal cord dorsal horn pain signaling [197], and there is growing evidence that astrocytic BDNF has an important role in nociception [152], chiefly in neuropathic pain, primarily because of the activation of trkB.T1 [198] (Figure 3C). Deletion of trkB.T1 reduces spinal cord injury-related abnormalities, including exaggerated responses to pain [199]. Indeed, trkB.T1 controls astrocytic proliferation, migration, and inflammation, and its deletion reduces their reactivity [200], supporting the idea that these cells depend on trkB isoform balance to regulate their physiological signaling output [195]. It is of note that trkB.T1 deletion reduces the behavioral response to capsaicin, a powerful TRPV1 agonist [198]. Astrocytic TRPV1, in turn, mediates astrocyte activation [201].

Other studies have demonstrated that BDNF is upregulated in cortical astrocytes under different pain conditions. Upregulation of the neurotrophin was observed in the ACC and S1 of rats with inflammatory pain [74]. Injections of recombinant BDNF (into the ACC) or a viral vector synthesizing BDNF (into the ACC or S1) trigger an elevation of neuronal LTP and sustained pain hypersensitivity, which are abolished after blocking trkB. In a different study, CFA injection induces BDNF expression in astrocytes (and neurons but not microglia) in Au1 [76].

Oligodendroglia

Oligodendrocytes serve as the myelinating cells in the CNS. They are the result of a cell lineage that must undergo a complex and perfectly timed process of growth, migration, specialization, and formation of the myelin sheet. Oligodendrocytes are considered to be highly susceptible cells in the CNS because of their unique metabolism and physiology, as well as the elaborate development program they undergo [202]. After spinal cord injury, oligodendrocytes produce BDNF [161] (Figure 3D). The contribution of oligodendrocytes to chronic pain is not well understood; nonetheless, some investigations have shown that their involvement in pain is caused by neurodegenerative, viral, or neuropathic conditions [203]. Even so, it was demonstrated that oligodendrocytes, by secreting BDNF, regulate the release of neurotransmitters at presynaptic brainstem terminals expressing trkB [153]. Therefore, it may be possible that some new links between this type of glia, BDNF, and pain will be discovered in the future.

#### 2.3.2. Immune Cells

Immune cells interact with neurons to modify pain sensitivity and facilitate the shift from acute to chronic pain [204]. T cells, B cells, and macrophages can produce circulating BDNF, especially in cases of inflammation and injury [205]. Strong evidence has accumulated that the immune system has a role in neuropathic pain as a consequence of nerve injury, as authoritatively reviewed in the recent past [155]. The mediators causing neuropathic pain have been classified into three main groups, primary, secondary, and tertiary [206]. Primary mediators are released by the immune cells after injury and act peripherally on DRG neurons; secondary mediators are released in the spinal cord and functionally modulate microglia and astrocytes; and tertiary mediators, among which is BDNF, are released by the spinal glia and act on the central terminals of PSNs and the dorsal horn neurons. However, data are accumulating to indicate that the BDNF released at the periphery by certain types of immune cells has a role in chronic (neuropathic) and inflammatory pain. When BDNF was measured in parallel with different subpopulations of immune cells in patients with chronic pain and healthy controls, it was observed that serum levels of BDNF are lower in pain-suffering patients who display a reduction in the number of CD4+ CD8+ T cells, CD3- CD56_bright_ NK cells, and CD20+ CD3- cells [207]. Likewise, proBDNF was shown to be upregulated in inflammatory cells during arthrosynovitis [120], and, similarly, upregulation of BDNF in mast cells was observed following back pain associated with intervertebral disk degeneration [208], and intradiscal inflammatory stimulation induces spinal pain behavior and intervertebral disc degeneration in vivo with an increase in macrophage markers and BDNF (as well as other nociceptive markers) [209]. Furthermore, studies have shown that following various forms of severe sciatic nerve damage, resident macrophages expressing the Iba1 protein gather around DRG neurons, thereby exacerbating pain hypersensitivity caused by nerve injury [210]. This study also found that by inhibiting the expression of Iba1 and modifying the characteristics of resident macrophages, specifically shifting them from an M1 to an M2 phenotype, analgesic effects can be achieved in neuropathic pain caused by SNL. This is accompanied by an enhanced release of BDNF, which triggers the regeneration of the peripheral processes of the adult DRG neurons. Another report was supportive of the intervention of BDNF produced by peripheral macrophages in the genesis of neuropathic pain [211]. In this paper, after putting contact between the nucleus pulposus of the intervertebral disc and the sciatic nerve, several neuropathic behaviors were observed in mice of both sexes in the absence of activation of the spinal microglia, and pain hypersensitivity was barred by genetically disrupting BDNF electively in macrophages.

### 2.4. BDNF Polymorphisms Related to Pain Signaling

Several studies highlight the importance of genetic polymorphisms in the control of pain signaling [212,213]. The single nucleotide polymorphism Val66Met (rs6265) in the BDNF gene, which involves replacing valine with methionine, is of relevance to the present discussion since it has the potential to disrupt the control of the BDNF/trkB signaling pathway (see [214] for a recent review).

The BDNF Val66Met polymorphism leads to the inability of BDNF to mature correctly in secretory vesicles following its synthesis (Figure 1). Studies in hippocampal neurons [215] have shown that Met-BDNF is primarily found in the cell body, but Val-BDNF is also present in distal dendritic processes. Furthermore, whereas Val-BDNF is the predominant form found in the medium from cultured neurons, the levels of Met-BDNF were hardly detectable. This indicates a difference in the release of the two forms of BDNF. It is worth remembering here that the Met allele is associated with a decrease in the sorting of proBDNF and reduced secretion of BDNF when neurons are depolarized [216], likely because of decreased interaction with sortilin [215]. The aberrant interaction between mutant BDNF and the sortilin or SorCS2 may not only hinder the release of BDNF but also cause deficiencies in neuronal development and plasticity [118].

In their very insightful study, Tian and colleagues found that individuals who possess the G allele (Val/Met and Val/Val) are at greater susceptibility to chronic postsurgical pain in comparison to those with the A allele of rs6265 [217]. These authors also provided evidence that *Bdnf* Met/Met mice have reduced mechanic allodynia following a plantar incision compared to the *Bdnf* Val/Val group. In terms of pain perception, the Met allele may have a dual function, acting as an analgesic under normal circumstances and as a promoter of pain sensitivity under chronic pain. Indeed, the Met allele has been shown to increase the sense of discomfort during chronic pain, compared to Val homozygotes [218]. The Val66Met polymorphism may have a direct effect on gene methylation, and the resulting expression and data show that individuals with chronic fatigue syndrome and fibromyalgia have higher amounts of BDNF in their bloodstream. These elevated levels of BDNF are directly related to the severity of pain sensations experienced by these patients, as compared to healthy individuals [219].

It was also hypothesized that the presence of the BDNF Val66Met variant might combine with early exposure to pain-related stress in preterm children, leading to distinct effects on the regulation of the hypothalamic-pituitary-adrenal axis. In an investigation of these issues, a correlation between reduced BDNF availability (indicated by the presence of the Met allele) and susceptibility to newborn pain and stress was observed in boys but not in girls [220]. In keeping with these observations, an elevated stress response is consistently found in mice with the BDNFMet knock-in mutation, in parallel with notable alterations in BDNF levels and dendritic spine density in the PFC and AMYG that are involved in the processing of the emotional component of pain [221].

Several very recent clinical studies have found a correlation between BDNF polymorphism and inflammatory pain. Compared to non-carriers, subjects suffering from osteoarthritis with BDNF polymorphisms are less likely to improve after rehabilitation [222] and, at EEG, display decreased alpha oscillations in the frontal area, likely reflecting the disruption of resting states to also compensate for the increased injury associated with joint inflammation [223]. Other investigations have disclosed an association between BDNF polymorphism and fibromyalgia (reviewed in [224]), and a systematic review and meta-analysis study demonstrated that the circulating BDNF levels are significantly higher in patients suffering from fibromyalgia than in controls [225]. Other studies examined the role of BDNF genotype and expression in the development of chronic low back pain. The findings of one of these studies suggest that having one or more minor alleles, which are associated with decreased BDNF activity, may lower the risk of transitioning from acute to chronic low back pain, possibly as a consequence of a decrease in BDNF signaling-dependent neuropathic pain mechanisms [226]. However, other authors subsequently failed to find an association between the Val66Met polymorphism of the BDNF gene with clinical and biopsychosocial characteristics of chronic low back pain [227]. It was also reported that BDNF polymorphism is associated with neuropathic pain in female cancer survivors [228] or peripheral neuropathy in type II diabetic patients [229].

## 3. BDNF and Neuronal Sensitization

Much of the above-described effects of BDNF may be linked to neuronal sensitization. There are two forms of neuronal sensitization, peripheral and central. After an accident or tissue damage, the lesioned tissue becomes sensitized to facilitate recovery by triggering behaviors that would avoid additional injury, such as guarding. Both the CNS and PNS have a role in sensitizing the nociceptive system, resulting in primary hyperalgesia in the periphery, secondary hyperalgesia, and allodynia in the CNS.

The presence and duration of the so-called hyperalgesic priming is another significant matter in pain plasticity. Priming occurs as a result of an initial injury and leads to a notable vulnerability to often insignificant painful stimuli, causing a prolonged state of pain in animals that have been primed [230]. Hyperalgesic priming can occur at the level of the nociceptors and relies on the activation of PKCε after local translation. It also can arise in the CNS, i.e., the dorsal horn of the spinal cord (and although little investigated the SNTN) where it depends on an atypical isoform of PKC (aPKC) referred to as PKMζ (reviewed in [230]). Interestingly, BDNF participates in the regulation and maintenance of hyperalgesic priming in both the PNS and CNS [231].

### 3.1. Peripheral Sensitization

Peripheral sensitization is the sensitization of the nociceptive system in the PNS. It is characterized by increased responsiveness and reduced threshold of nociceptive neurons to stimulation of their receptive fields. The increased responsiveness of the nociceptors is linked to an increase in the expression, sensitivity, and receptiveness of receptors such as the TRP channels TRPV1 and transient receptor potential cation channel, subfamily A, member 1 (TRPA1) [232], which participate in the hypersensitivity of PSNs to mediators such as CGRP, substance P, and BDNF [27]. BDNF, together with other neurotrophins and glial-derived neurotrophic factor (GDNF), is an important modulator of intraepidermal nerve fibers and is involved in the regulation of pain and itch [233]. BDNF enhances the sensitivity of peripheral nociceptors, making them more responsive to painful stimuli. This effect is mediated, among others, by the upregulation of ion channels, such as TRP channels and voltage-gated Na^+^ channels, in sensory neurons [78]. Peripheral sensitization of nociceptors can also depend on the BDNF released by the Schwann cells activating the pronociceptive PKCε [157] (Figure 3A). Notably, peripheral inflammation leads to an upregulation of BDNF mRNA and protein in trkA-positive PSNs (which normally do not produce BDNF) via an NGF-dependent mechanism [234], the so-called phenotypic switch that occurs in both inflammatory [86] and neuropathic [235] pain. Nociceptor sensitization is believed to be sustained by elevated peripheral levels of BDNF and its anterograde transport to dorsal horn neurons. It was shown that peripheral inflammation and the electrical activity in C-fibers result in an upregulation of BDNF expression in the DRG neurons [236]. This increase also leads to increased trkB in the dorsal horn [123]. The augmented production and occurrence of BDNF in PSNs, as well as its transport along the axons to the terminals of their peripheral branches, have also been linked to the regulation of acute and inflammatory pain [99,237]. As mentioned, increased production of BDNF in DRG neurons may be influenced by a higher concentration of NGF [147,238,239] and will result in a greater amount of the factor being present at the periphery. Interestingly, a considerably increased release of BDNF was observed when acute nociceptor stimulation was applied to the inflamed human skin [240]. Given that elevated levels of BDNF necessitate an enhanced buildup of peripheral neurotrophins, these findings indicate an increased retrograde axonal transport of BDNF from the skin to the cell bodies of PSNs.

Another effect of nerve injury is the synthesis of macrophage CSF-1 in primary afferents and its release in the dorsal horn [173]. CSF-1 increases the excitatory drive to excitatory dorsal horn neurons via BDNF activation of presynaptic and postsynaptic trkB and p75^NTR^. In parallel, CSF-1 reduces the stimulation of inhibitory neurons by excitatory signals through mechanisms that do not involve BDNF.

### 3.2. Central Sensitization

BDNF contributes to central sensitization, a process that amplifies pain signals in the CNS [241]. In central sensitization, CNS neurons begin generating action potentials even in the absence of stimuli from nociceptors due to changes within the neurons themselves. Central sensitization can arise through the rearrangement of synapses in the spinal cord and terminal sensory nuclei of the cranial nerves following prolonged nociceptor activation. This can result in the inappropriate transmission of pain signals to other CNS areas. The rearrangement of synapses is commonly referred to as synaptic plasticity, and BDNF plays an important role in this process [83]. LTP, perhaps the most investigated form of synaptic plasticity, is a well-known process in the CNS where there is a permanent enhancement in the effectiveness of communication between neurons. LTP at C-fiber synapses in the spinal dorsal horn and SNTN is regarded as a synaptic representation of pathological pain [242]. This is because spinal LTP is specifically triggered by nociceptive but not innocuous stimuli. Additionally, stimulation that induces LTP can cause persistent behavioral indications of pathological pain in both humans and animals. The role of BDNF in spinal plasticity has been authoritatively reviewed [83]. The chemical mechanisms underlying LTP at C-fiber synapses in the spinal cord (Figure 2) share similarities with LTP in the hippocampus in several respects. The induction of LTP relies on an increase in Ca^2+^ in the postsynaptic SSN. This increase is caused by the opening of NMDA channels and voltage-gated Ca^2+^ channels, as well as the release of Ca^2+^ from intracellular stores. Early-phase LTP requires the activation of intracellular PKA, PKC, CaMKII, PLC, and the production of nitric oxide (NO) [242]. It was demonstrated that inhibition of PLCγ-PKC signaling blocks the enhancement of synaptic responses in lamina II caused by BDNF, specifically in response to C-fiber stimulation [80]. Activation of the PLCγ pathway leads to increased [Ca^2+^]_i_ levels, which can happen by releasing Ca^2+^ either from internal stores or across phosphorylated calcium-permeable glutamate receptors. These findings are consistent with multiple studies that suggest that the activation of MAPK/ERK and PLCγ-PKC, along with the increase in [Ca^2+^]_i_, play crucial roles in the development of nociceptive plasticity, inflammatory, and neuropathic pain. The LTP of C-fiber responses is enhanced by BDNF, and BDNF pretreatment prevents subsequent LTP generated by SNL [243]. The significance of these effects is emphasized by the discovery that antibodies targeting CSF-1 can block spinal LTP, microglial activation, and induction of BDNF [173].

LTP that occurs after more than three hours (late-phase LTP) is reliant on the synthesis of new proteins. The induction of late-phase LTP can be achieved by activating either dopamine D1 receptors or PKA, as well as by administering exogenous BDNF or ATP directly. Consequently, drugs that target these molecules may negatively affect the memory function of the hippocampus. The notable distinction between hippocampal LTP and spinal LTP at C-fiber synapses lies in the fact that the activation of glial cells and the excessive production of proinflammatory cytokines, such as TNF-α and IL-1β, impede LTP in the hippocampus, whereas they facilitate LTP in the spinal dorsal horn [242].

LTP at C-fibers causes hyperalgesia and is activity-dependent (reviewed in [244]). It can be induced by high-frequency electrical stimulation or conditioning low-frequency stimulation but also by many of the conditions described in the previous sections to model inflammatory or neuropathic pain, i.e., noxious thermal, chemical (formalin, capsaicin), or mechanical (pinching) stimulation of the skin. Induction of LTP varies among afferent fibers innervating different organs or tissues. LTP is more difficult to induce in nociceptive skin afferents than in nociceptive muscle afferents [245]. This difference has been correlated to the relative lack of BDNF in cutaneous afferents, giving further strength to the importance of the molecule in central sensitization. As a result of the binding of BDNF to trkB, activation of the PI3K-Akt, PLCγ, and MAPK/ERK pathways and phosphorylation of the NMDAR1 subunit occurs in target neurons [246].

It is of relevance that BDNF was demonstrated to play a crucial role in protein synthesis-dependent long-term LTP in the spinal dorsal horn via activation of ERK, p38 MAPK, and NF-kB signal pathways [247]. In the pathogenesis of several types of inflammatory or neuropathic pain, the NF-kB pathway is also implicated in upregulating BDNF after activation of the proteinase-activated receptor 2 (PAR2) [248]. In addition, spinal application of BDNF in rats has been demonstrated to contribute to spinal late-phase LTP and mechanical hypersensitivity by activation of microglia, Src-family kinases, and p38 MAPK in the absence of any C-fiber activation [249].

Interestingly, it was suggested that the initial processes of central sensitization involve the release of BDNF, most likely originating from the neurons. This is then followed by a short period of elevated expression of BDNF and phosphorylated trkB in glial cells, which may trigger the transition from early-phase sensitization to late-phase sensitization [250]. Observations on the transgenic mouse L1-DREAM that displays reduced expression in the spinal cords of several genes related to pain, including BDNF, are consistent with the notion that neuronal BDNF has a primary role in central sensitization [251]. The downstream regulatory element antagonist modulator (DREAM) modulates the endogenous responses to pain, and this study found that peripheral inflammation causes an increase in spinal reflexes and the production of BDNF in wild-type mice but not in DREAM transgenic mice. In vitro, the amplification of the spinal reflexes was replicated with continuous electrical stimulation of C-fibers in preparations from wild-type but not in transgenic mice. Finally, transgenic mice display a lasting improvement in their dorsal root-ventral root responses because of exogenous BDNF. Thus, BDNF contributes to the heightened sensitivity of the spinal cord following inflammation. Conversely, its absence in DREAM transgenic mice is the cause of the inability of these animals to develop central sensitization in the spinal cord.

The role of BDNF in central sensitization after SCI is however quite elusive, as it has been suggested that SCI modifies the influence of BDNF on the activity of the GABAergic neurons. After SCI, the administration of capsaicin to a single hind paw results in heightened sensitivity to mechanical stimulation that can be averted by delivering a regular (fixed-spaced) stimulation before the injection of the vanilloid [252]. The protective effect is nullified when rats are pretreated with neutralizing trkB-IgG antibodies. BDNF decreases ERK and pERK after SCI, displaying opposing effects than those observed in non-injured rats. These divergent effects are linked to different impacts on GABAergic neurons. In non-injured rats, BDNF reduces the expression of KCC2 and attenuates the inhibitory effect of the GABA_A_R agonist muscimol, whereas, after SCI, BDNF causes an upregulation of KCC2, which would enhance the recovery of GABAergic inhibition.

In another model of neuropathic pain, SNL, it was investigated whether BDNF controls the activation of GluN2B-NMDARs through Src homology-2 domain-containing protein tyrosine phosphatase-2 (SHP2) phosphorylation to produce pain [133]. It was thus established that spinal BDNF is involved in the central sensitization of dorsal horn wide dynamic range (WDR) neurons and the occurrence of pain allodynia in both intact and SNL rats. Furthermore, it was discovered that BDNF triggers LTP in C-fiber synapses by enhancing the activity of GluN2B-NMDARs in the spinal dorsal horn. Finally, it was confirmed that the phosphorylation of SHP2 by BDNF is necessary for the subsequent up-regulation of GluN2B-NMDARs and the induction of spinal LTP with the development of allodynia.

### 3.3. Hyperalgesic Priming in the Spinal Cord Dorsal Horn

As mentioned, BDNF interacts with aPKCs in the beginning and maintenance of hyperalgesic priming in the CNS [253]. Activation of nociceptors triggers the release of BDNF in the spinal cord, which, in turn, leads to the translation of aPKC through the activation of trkB and the involvement of the mammalian target of rapamycin complex 1 (mTORC1). Elevated levels and phosphorylation of aPKCs are believed to have a role in starting priming. After priming is established, the elevated levels of aPKC protein and phosphorylation result in a continuous augmentation of AMPAR transportation to the postsynaptic membrane with a persistent augmentation of postsynaptic glutamate-mediated signaling [254]. BDNF signaling through trkB is likely responsible for the regulation of this process. Hyperalgesic priming, once established, can be irreversibly reversed through the inhibition of aPKC using Zrt, Irt-like proteins (ZIP), the disruption of AMPAR trafficking with a peptide called pep2M, or the inhibition of trkB/BDNF signaling using ANA-12 or trkB-Fc, respectively.

## 4. Implications for Pain Management

Understanding the role of BDNF in nociception has significant implications for the management of chronic pain and pain-related conditions. There are three different therapeutic strategies for pain treatment derived from preclinical research through inhibition of BDNF/trkB signaling: the sequestration of BDNF, the blocking of the extracellular trkB region, or disruption of the intracellular kinase region of trkB. These approaches have therapeutic potential but are far from being of use in clinical settings.

To scavenge BDNF in living organisms, one can use a trkB-Fc fusion protein derived from the extracellular domain of trkB and the Fc domain of human IgGs [255]. This method has been widely employed to investigate the function of naturally occurring BDNF in various pain models, e.g., [148,256,257]. From the studies on these models, it was demonstrated that blocking neuronal or glial BDNF signaling in vitro and/or in vivo impacts the onset of pain-related behaviors and the connected cellular signaling. While the approach is highly targeted, its applicability in clinical settings is limited due to the requirement of intrathecal or local application at the exact central areas where BDNF is released.

To decrease the likelihood of ligand binding, one can target the extracellular domain by using trkB-blocking antibodies [258] or by synthesizing novel receptor antagonists [259]. Monoclonal antibodies that block the function of trkB have been used successfully in preclinical studies. Specifically, a mouse monoclonal antibody [260] has been shown to effectively block the effects of BDNF on neuronal activity in acute spinal cord slices [56]. Additionally, other neutralizing antibodies have been found to reverse neuropathic [148] and other types of pathological pain [261] in rodents. Like trkB-Fc, anti-trkB blocking antibodies that are delivered systemically do not cross the blood-brain barrier (BBB) and thus cannot penetrate the CNS. This, combined with their relatively limited sensitivity, makes them unsuitable for clinical therapy.

Over the past ten years, the discovery of small new compounds that function as negative allosteric modulators of trkB has led to significant progress in understanding how receptor ligands are processed in the body and their ability to be absorbed into the bloodstream. A peptidomimetic strategy was used to discover cyclotraxin-B, a tiny fragment of BDNF. This fragment can modify the conformation of trkB through an allosteric mechanism [262]. Subsequently, cyclotraxin-B was demonstrated to possess distinct antinociceptive properties in various pain models [74,154,263,264,265,266] to be able to block LTP and central sensitization induced by BDNF [250], as well as the in vitro activation of PKCε in rat Schwann cells that, as mentioned, plays a role in peripheral sensitization [267] (see Section 3.1 Peripheral Sensitization). However, the peptide has little specificity, as it affects both BDNF-dependent and -independent trkB activation. In addition, cyclotraxin-B demonstrates efficacy when given intravenously but not by other tested routes of administration. These shortcomings were bypassed after discovering ANA-12, a non-peptidic small molecule that allosterically inhibits the binding of BDNF to trkB at nanomolar concentrations [259]. ANA-12 has shown great potential in preclinical research, with a growing body of evidence confirming its effectiveness in reducing pain in many experimental settings [90,115,183,186,268,269,270,271,272,273,274,275,276,277,278,279,280,281,282,283]. Unlike cyclotraxin-B, ANA-12 can be administered orally, making it a potential and targeted option for clinical trials.

To inhibit the activity of tyrosine kinases within cells, the primary method is to disrupt the ATP binding site of these enzymes [284,285]. K252a, an indolocarbazole molecule, is among the initial substances employed as a competitive inhibitor of the ATP site, effectively obstructing trk catalytic activity [285,286]. Administration of K252a successfully inhibits the increase of intracellular Ca^2+^ induced by BDNF in acute slices of the rat superficial dorsal horn [56]. Moreover, in vivo, it diminishes hypersensitivity in several pain models [148,287,288,289,290,291,292,293,294,295]. Based on promising results from preclinical studies, various pharmaceutical companies have allocated significant resources toward developing and ameliorating kinase inhibitors. Many of these inhibitors have been patented, and some are (or have been) undergoing clinical trials or investigations for a wide range of diseases and pain [284,296,297,298,299]. While several of these compounds demonstrate significant effectiveness in laboratory tests, the creation of a trkB inhibitor that can be successfully brought to market has not yet been achieved. Regrettably, the task of creating tyrosine kinase antagonists for individual trks is difficult because ATP competitive trk inhibitors, at most, are only selective to trk receptors as a whole (i.e., they are pan-trk inhibitors) and fail to discern between trk subtypes [284]. Hence, a significant drawback associated with the utilization of these compounds is the potential for developing adverse CNS effects. To minimize negative consequences, numerous endeavors have been undertaken to create compounds that do not penetrate the BBB but mostly exert their actions at the periphery. Certain compounds, such as ARRY-470 [300] or PF-06273340A [301], demonstrate antinociceptive effects in animal models of chronic pain. The efficacy of the pan-trk inhibitor ONO-4474 as an analgesic in patients with moderate to severe osteoarthritis was demonstrated in a recent randomized, double-blinded clinical trial [302]. However, it seems very likely that at the periphery, these chemicals also affect other trk receptors in addition to trkB, such as, e.g., trkA.

As a result of the restrictions mentioned above, there are currently no successful pain therapies that rely on BDNF/trkB. The crucial function performed by BDNF and trkB in the preservation of central neurons imposes further constraints on the utilization of trkB antagonists as a secure treatment strategy. Furthermore, the intricate and diverse nature of the intracellular signaling cascades triggered by tyrosine kinase receptors poses challenges in the development of targeted pharmaceutical interventions. Targeting BDNF/trkB might result in unintended side effects caused by the deactivation of BDNF-dependent pathways crucial for the proper functioning of healthy neurons or by the lack of specificity and interaction with other non-BDNF pathways [303].

To tackle trkB-dependent pathological changes, alternative treatment approaches can be pursued by targeting either the upstream or downstream effectors. Targeting upstream microglia can effectively decrease the BDNF-induced changes in neuropathic pain [172]. Microglial P2RX4, when activated, plays a crucial role in releasing BDNF. Therefore, targeting these receptors could be a promising approach for clinical therapies [304]. As mentioned, one important way that trkB-dependent pain hypersensitivity occurs is through the downregulation of KCC2, which leads to an imbalance between excitatory and inhibitory neurotransmission in downstream signaling pathways [148]. Therefore, KCC2 is considered a potential pharmaceutical target [305], and the making of KCC2 enhancers holds great potential as a therapeutic approach to reinstate inhibition and mitigate the symptoms linked to the BDNF-trkB-KCC2 cascade [306].

## 5. Clinical Trials

In this section, I will briefly consider the clinical trials of the last ten years related to BDNF in the context of pain modulation. It is worth noting that these trials have focused on measuring BDNF as a biomarker in several clinical pain conditions of a heterogeneous nature rather than trying to translate into the clinic the results of the preclinical studies that have used the approaches described in the previous section to inhibit BDNF/trkB signaling.

Among the pathologies, one of the most intensely investigated is fibromyalgia, a persistent and recurring disorder characterized by widespread pain that is accompanied by intense emotional discomfort and functional impairment. Several clinical trials were devoted to the study of BDNF levels in fibromyalgia because people diagnosed with this condition exhibit elevated levels of serum BDNF compared to healthy persons, suggesting a significant involvement of BDNF in the pathophysiology of the disorder [307]. One of the first trials using serum BDNF as a biomarker dates back to 2014 and aimed, among others, at assessing the association with conditional pain modulation (CPM) [308]. In this randomized, double-dummy, placebo-controlled study, the cold-heat task (CPM-TASK) was used as an experimental pain stimulus to activate the diffuse noxious inhibitory control-like effect for evaluating the effect of melatonin analgesia. Mean serum BDNF after treatments with amitriptyline, melatonin, or melatonin+amitriptyline is reduced by 22.57%, 36.6%, and 34.49%, respectively, compared to baseline. However, delta values (serum BDNF before treatment minus serum BDNF after treatment) show a lack of statistically significant differences. It was thus suggested that BDNF could be used as a biomarker of central sensitization and correlated with pain reduction in the CPM-TASK, although with some caveats. In another study, plasma samples were analyzed after a 15-week progressive resistance exercise to show that the levels of NGF are unchanged while those of BDNF increase, suggesting that BDNF may affect nociception/pain in fibromyalgia [309]. In addition, serum BDNF was proposed to be a valuable predictor of the tDCS effect on pain score decreases across the treatment for the condition [310,311]. In keeping with these observations are those in patients not only suffering from fibromyalgia but also other forms of central sensitivity syndrome such as osteoarthritis and endometriosis, leading to somatic and visceral pain, respectively [312]. That serum BDNF inversely correlates with different forms of postoperative pain was also demonstrated using a new predictive tool for postoperative pain, the brief measure of emotional preoperative stress (B-MEPS), to find that blood BDNF is inversely correlated with morphine consumption and length of stay after surgery [313]. Likewise, serum levels of BDNF and trkB are inversely associated with depressive symptoms and sleep quality in patients undergoing adjuvant chemotherapy for breast cancer [314].

The effect of repetitive transcranial magnetic stimulation (rTMS) in chronic myofascial pain syndrome was investigated in a double-blinded, randomized, sham-controlled trial [315]. It was observed that rTMS reduces pain scores in parallel with an increase in serum BDNF. The increase in serum BDNF in the rTMS-treated patients was interpreted as an indication of the neuroplasticity that underlies the therapeutic effect of rTMS, and serum BDNF was considered as a surrogate marker that could be used to monitor the therapeutic effects of rTMS rather than a bona fide marker of the pain severity. These authors have also investigated the effect of electroacupuncture in chronic tension-type headache (CTTH) in a randomized, sham-controlled, crossover trial and concluded that electroacupuncture analgesia in CTTH is related to neuroplasticity that can be monitored by serum BDNF [316]. Partly in agreement with these observations, another group observed a decrease in urinary NGF but not BDNF in patients with interstitial cystitis/bladder pain syndrome treated with hyaluronic acid [317]. Moreover, in a randomized, double-blind, factorial design and controlled placebo-sham clinical trial using rTMS and deep intramuscular stimulation therapy in chronic myofascial pain syndrome, no variations in blood BDNF were observed, although both treatments are effective in relieving pain [318]. The variation of serum BDNF levels following integrated multimodal intervention in postherpetic neuralgia was studied in a randomized, double-blind controlled study to conclude that minimally invasive pulsed radiofrequency and pregabalin are effective in early pain reduction that is accompanied by elevated serum BDNF levels [319]. In another randomized, double-blind, controlled study to test the effectiveness of intraoperative ketamine on postoperative depressed mood after elective orthopedic surgery, there was an increase in blood BDNF after surgery [320]. Similar results were obtained in a randomized sham-controlled study on patients with persistent chronic pain after hallux valgus surgery where an increase in the level of BDNF in cerebrospinal fluid was observed after anodal transcranial direct current stimulation (tDCS) and was again interpreted as a result of an indirect measurement of neuroplasticity changes induced by tDCS [321]. Similarly, another group observed that serum BDNF increases after three weeks of high-frequency rTMS in patients with SCI [322]. These authors speculated that the frequency of rTMS may play an important role in the BDNF secretion processes and, in keeping with most of the findings reported above, showed a negative correlation between serum BDNF levels and pain scores. Transcutaneous electrical nerve stimulation (TENS) has been used to relieve osteoarthritic knee pain, and a clinical trial has explored the effects of genotype on TENS efficacy without finding any correlation with the expression of the *BDNF* gene [323].

Altogether, a large part of these and other studies [324,325] converged to the conclusion that BDNF could be a good serum biomarker of neuroplasticity state under different clinical settings related to a pain experience with an increase in serum BDNF in parallel with the reduction of pain. If this interpretation is correct, one should consider that the elevation of BDNF is related to its function as a trophic factor rather than a pain modulator. On the other hand, in a study on fibromyalgia, attachment-based compassion therapy seemed to reduce serum BDNF and appeared to be correlated to anti-inflammatory effects on patients, leading to speculation that reduction in BDNF could be a mechanism of functional status improvement [326].

At the beginning of this year, a study protocol for a randomized clinical trial was proposed, aiming to test whether a targeted biobehavioral therapeutic education program could induce alterations in pain perception and biomarkers of brain plasticity (among them BDNF) in individuals suffering from chronic pain [327]. Notably, the authors emphasized that it is crucial to acknowledge that measuring BDNF levels can be challenging because of the inherent unpredictability of collecting, storing, and analyzing samples. Their study will focus on measuring BDNF levels in plasma samples since they have been determined to be more resistant to changes during analysis compared to serum samples [328], which were, however, the most used in the studies quoted above. The investigation will be performed with standardized ELISA kits, incorporating regular calibration and quality control checks to mitigate potential sources of variation. For future studies, implementing these steps is essential to guarantee the precision and authenticity of the BDNF measurements and maintain the integrity and dependability of the study’s results.

## 6. Conclusions

BDNF plays a significant role in the modulation of nociception and pain signaling. Its effects on peripheral nociceptors, synaptic transmission, and central sensitization are key aspects of pain processing. Understanding the molecular mechanisms behind BDNF’s role in nociception provides valuable insights into potential targets for pain management therapies, offering hope for improved treatments for pain-related conditions. Future research should also be directed at understanding the interactions of BDNF with other pain modulators. GDNF, for example, has antinociceptive opposite effects to BDNF and is present with BDNF in the spinal cord dorsal horn. The circuitry of the interactions of the two molecules in the substantia gelatinosa (lamina II) is starting to be unveiled [23,53,329,330,331] and could offer new opportunities to develop more targeted pain therapies.

Box 1Overview of the Anatomical Arrangement of Pain Pathways  The somatic pain pathways [332] are responsible for gathering sensory input from several sources, including the skin, muscles, joints, ligaments, and bones. Except for the head and the proximal regions of the neck, which receive sensory innervation from the peripheral projections of the trigeminal PSNs, the transmission of pain signals from all other body parts is initially carried out by the PSNs of the dorsal root ganglia (DRGs). From these two sets of neurons originate the main pathways responsible for transmitting nociceptive signals to the brain: the trigeminothalamic (TTP) and spinothalamic (STP) pathways. Both are composed of a polysynaptic chain including three sensory neurons, commonly known as first-, second-, and third-order somatic sensory neurons. The sensory discriminative features of pain perception are reliant upon the crucial role of these cells. The first-order sensory neurons, i.e., the PSNs, are situated in the trigeminal ganglion
(TG), the proximal ganglia of the glossopharyngeal (petrosal ganglion,
PG), and vagus nerves (nodose ganglion, NG), or the DRGs. The SSNs are
located within the dorsal horn of the spinal cord or the spinal nucleus of the trigeminal nerve (SNTN). These
are projection neurons that give rise to long axons crossing the midline and
ascending throughout the spinal cord and/or brainstem, ultimately reaching
the thalamus. Third-order sensory neurons are in the ventroposterior lateral nucleus of the thalamus (VPLN)
and project their axons to the first-order (S1) and second-order (S2)
somatosensory areas of the cerebral cortex. Nociceptive signal modulation can
take place at various points within this polyneuronal chain. It is primarily
observed at the substantia gelatinosa of the dorsal horn (lamina II) of the
spinal cord or the SNTN. Additionally, several descending pathways can exert
inhibitory or facilitatory effects on the TTP or STP neurons.  Other ascending pathways play a significant role in
the overall perception of pain. These pathways transmit stimuli associated
with motivational and cognitive aspects (the spinoreticular tract and
spinoparabrachial tract), motor responses, and affectivity (the
spinomesencephalic tract and spinoparabrachial tract), as well as
neuroendocrine and autonomic responses (the spinohypothalamic tract).
Remarkably, the spinoparabrachial pathway (SPBP) serves
as a significant site of convergence for both somatic and visceral
nociceptive stimuli. It is also worth noting that besides nociceptive
responses, visual and auditory stimuli can generate exteroceptive responses
that contribute to pain [333].  These basic circuits may undergo modifications in
switching from physiological (acute) to pathological (chronic) pain.
Likewise, additional pathways may be recruited in several conditions leading
to chronic pain. The polysynaptic dorsal column
pathway (DCP) is, e.g., involved in the processing of pain following
peripheral nerve damage, as a phenotypic switch (see Section 3.1—Peripheral Sensitization) occurs
in DRG neurons projecting to the gracile and cuneate nucleus following nerve
injury [334,335,336].  Somatic pain stimuli that reach the thalamus relay
nuclei (VPLN and ventroposterior inferior nucleus) are primarily conveyed to
S1. However, other cortical areas are involved in the processing of these
stimuli, including S2. A ventrally oriented cortico-limbic somatosensory
circuit connects S1 and S2, integrating somatosensory input with learning and
memory as well as other sensory modalities such as vision and audition. S1
and S2 also connect with the posterior parietal cortical areas and the
insular cortex (INS). The INS is, in turn, linked to the AMYG, perirhinal
cortex, and hippocampus.  Visceral pain originates from the internal organs,
such as the heart, blood vessels, airways, gastrointestinal tract, and
urinary and reproductive organs [337].
Nevertheless, it should be noted that visceral discomfort does not arise from
all internal organs, nor is it necessarily associated with demonstrable
physical harm to these organs. The challenging task of identifying and
localizing visceral pain stems from the comparatively sparse distribution of
sensory nerves in visceral organs and the vast dispersion of visceral input
within the CNS. Input from the viscera has a broader and less precise
distribution, primarily targeting lamina I and the deep dorsal horn. In
different individuals, pain originating from various visceral organs can
exhibit distinct patterns of presentation, such as discomfort radiating from
the bladder to the perineal area or from the heart to the left arm and neck.
Consequently, visceral pain frequently exhibits localization to anatomical
sites that are distant from its actual source, leading to its designation as
referred pain. Thus, symptoms often involve the occurrence of referred pain
to somatic tissues located in the same metameric field as the damaged
viscera. Furthermore, it is worth noting that secondary hyperalgesia can
manifest in the superficial or deep body wall tissues because of
viscerosomatic convergence. Visceral discomfort is frequently accompanied by
notable motor and autonomic reactions, including excessive perspiration;
feelings of nausea, vomiting, and gastrointestinal disturbances; and
alterations in body temperature, blood pressure, and heart rate.  The afferent fibers that primarily innervate the
viscera transmit signals to the CNS via sympathetic and parasympathetic
neurons. The sympathetic innervation is provided by visceral afferent fibers
that originate from a small subset of DRG neurons and travel along the
hypogastric, lumbar, and splanchnic nerves, which traverse both prevertebral
and paravertebral ganglia to connect with their target organs. The
parasympathetic innervation is made of the afferent fibers originating from
the glossopharyngeal (IX), vagus (X), and pelvic nerves, which end in the
brainstem and lumbosacral cord, respectively. While it is true that vagal
afferents do not directly transmit pain signals, multiple investigations have
shown evidence that the stimulation of the vagus nerve can reduce both
somatic and visceral pain. The glossopharyngeal and vagal afferents arise
from the PSNs in the PG and NG, respectively, which then transmit signals to
the SSNs of the nucleus tractus solitarius (NTS)
in the medulla oblongata. Sympathetic visceral afferent converging in the
spinal cord synapse with SSNs located in the dorsal horn. These SSNs then
transmit signals to higher centers via the DCP, the SPBP, and a component of
the STP projecting to specific midline thalamic nuclei. The projections
originating from the superficial dorsal horn, primarily comprising the SPBP,
are linked to autonomic and emotional reactions to painful stimuli. In
addition to projections from vagal afferents, spinoparabrachial projections
are conveyed to higher relay centers associated with limbic and cognitive
functions, including brain regions implicated in affectivity, such as the
AMYG, hypothalamus (HYPO), and periaqueductal gray (PAG). The visceral
spinothalamic projections originate from the deep dorsal horn traverse to the
contralateral side and reach the ventroposterior medial
nucleus (VPMN) and VPLN of the thalamus. The medial thalamic nuclei
ultimately transmit signals to the cortical regions associated with visceral
pain. It is of relevance that the DCP also plays a part in visceral
nociception and that the gracile nucleus cells react to painful stimulation
of the viscera. This pathway originates from cells in lamina III of the
dorsal horn and is located slightly lateral to lamina X [335].  Visceral pain stimuli that are conveyed to the
midline thalamic nuclei are subsequently transferred to limbic cortical areas
such as the ACC and INS. Individual neurons often
project in more than one of these pathways. Significantly, the ascending
spinal connections directly target the limbic and subcortical areas where
this system ultimately converges. This twofold convergence could be
associated with a process wherein several different brain sources mediate the
impact of pain.

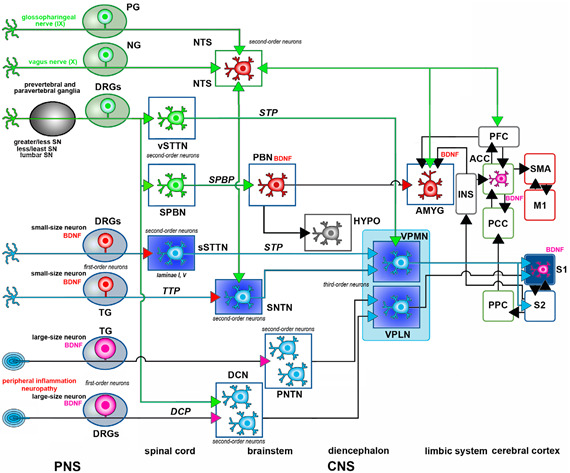

  **Box Figure**. Schematic representation of the
main neuronal pathways involved in nociception/pain. Somatic afferents are in
light blue, and visceral afferents are in green. BDNF-expressing neurons
under basal conditions are in red. Neurons expressing BDNF under
inflammation/neuropathy or chronic pain are in fuchsia. Abbreviations: ACC =
anterior cingulate cortex; AMYG = amygdala; BDNF = brain-derived neurotrophic
factor (protein); CNS = central nervous system; DCP = dorsal column pathway;
DCN = dorsal column nuclei; DRGs = dorsal root ganglia; HYPO = hypothalamus;
INS = insula; M1 = primary motor cortex; NG = nodose ganglion; NTS = nucleus
tractus solitarius; PBN = parabrachial nucleus; PCC = posterior cingulate
cortex; PFC = prefrontal cortex; PPC = posterior parietal complex; PG =
petrosal ganglion; PNS = peripheral nervous system; PNTN = pontine nucleus of
trigeminal nerve; S1 = primary somatosensory cortex; S2 = secondary
somatosensory cortex; SMA = supplementary motor area; SNTN = spinal nucleus
of trigeminal nerve; SPBN = spinal parabrachial neurons; SPBP =
spinoparabrachial pathway; STP = spinothalamic pathway; sSTTN = somatic
spinothalamic neurons; TG = trigeminal ganglion; TTP = trigeminothalamic
pathway; VPLN = ventroposterior lateral nucleus of the thalamus; VMLN =
ventroposterior medial nucleus of the thalamus; vSTTN = visceral
spinothalamic neurons.

## Figures and Tables

**Figure 3 biomolecules-14-00539-f003:**
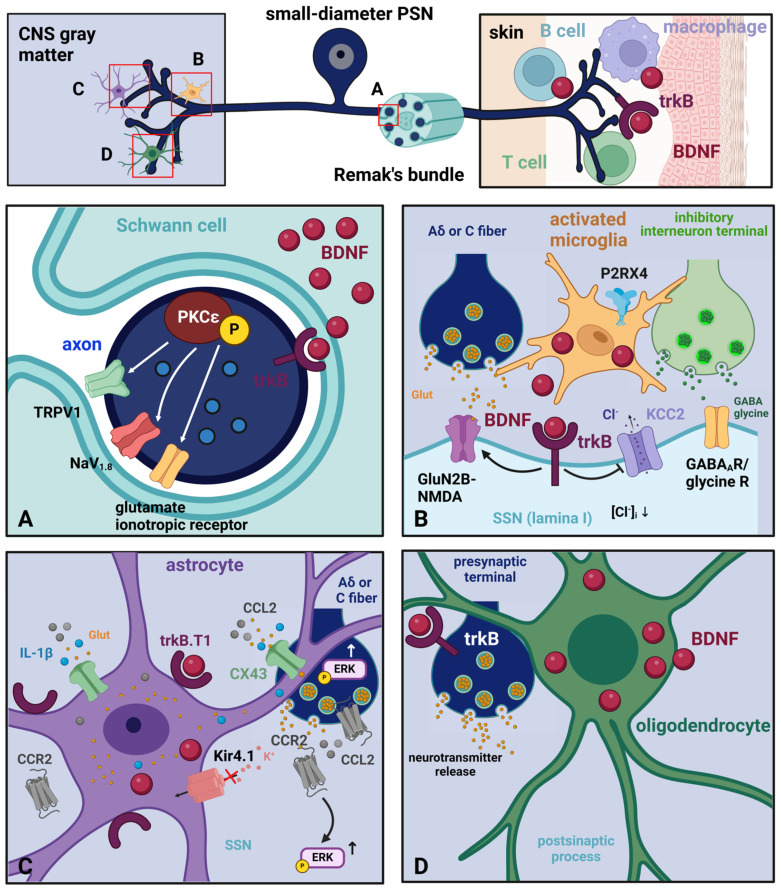
Glial and immune cell BDNF synthesis related to nociceptive/pain pathways. Top: a small-size pseudounipolar PSN giving rise to unmyelinated/poorly myelinated fibers of the Aδ-/C-type. These neurons have a single process that bifurcates to form a peripheral branch in tissues (e.g., skin) and a central branch reaching the CNS gray matter. At the periphery, the PSN terminals expressing trkB can bind BDNF released by immune cells. Areas delimited by the red squares are shown at higher magnification in the bottom panels to schematically depict the main mechanisms of action of glial BDNF. (**A**): Schwann cell released BDNF binds to trkB to phosphorylate PKCε in the peripheral branch of PSNs. PKCε, in turn, activates several ionotropic receptors in the PSN including glutamate receptors, TRVP1, and NaV_1.8_. (**B**): BDNF released from activated microglia promotes neuropathic pain. Following PNI, spinal microglia undergo activation and exhibit excessive expression of P2RX4. Under pathological circumstances, touch spikes elicited in Aβ fibers (not shown) lead to the release of GABA and ATP from inhibitory interneurons located in the dorsal horn. ATP that has been released activates P2X4Rs on microglial cells, leading to the release of BDNF. BDNF then acts on trkB of SSNs, resulting in the downregulation of KCC2 and an increase in [Cl−]_i_. Following these events, the released GABA impacts the Cl− channels of SSNs, causing an elevation in Cl− outflow, which, in turn, depolarizes these neurons and triggers action potentials. (**C**): Release of BDNF from astrocytes is dependent on trkB.T1. Under inflammation, astrocytes release glutamate and several bioactive molecules through CX43 channels, among which are IL-1β and CCL2. CCL2 acts on CCR2 to increase pERK in Aδ- or C-fiber terminals of PSNs and SSNs. Inhibition of Kir 4.1 induces BDNF expression in astrocytes. (**D**): Oligodendrocytes release BDNF after injury, and the neurotrophin acts on presynaptic trkB to promote neurotransmitter release. Abbreviations: BDNF = brain-derived neurotrophic factor; CCL2 = chemokine (C-C motif) ligand 2; [Cl−]_i_ = intracellular chloride concentration; CCR2 = C-C chemokine receptor type 2; ERK = extracellular signal-regulated kinase; GABA_A_R = γ amino-butyric acid receptor A; GLUN2B-NMDA = NMDA receptor GluN2B; GlyR = glycine receptor; KCC2 = K-Cl co-transporter 2; Kir 4.1 = inwardly rectifying potassium channel 4; NaV_1.8_ = voltage-gated sodium channel 1.8; PKCε = protein kinase Cε; PNI = peripheral nerve injury; P2RX4 = ATP-gated purinergic receptor 4; PSN = primary sensory neuron; SSN = secondary sensory neuron; trkB = tropomyosin receptor kinase B; trkB.T1 = truncated isoform 1 of trkB; TRPV1 = transient receptor potential vanilloid 1. Created with BioRender.com.

**Table 3 biomolecules-14-00539-t003:** Immunocytochemical distribution of BDNF in pain pathways glia.

Location	Type of Cells	Function	Ref
PNS	Schwann cells	Modulation of nociception	[157]
DH	Activated microglia (OX-42+/Iba1+)	Inflammatory/neuropathic pain	[158,159,160]
	Oligodendrocytes (APC+)	Response to injury	[161]
SNTN	Microglia (Iba1+)	Nociception (allodynia)	[115]
Au1	Astrocytes (GFAP+)	Chronic pain	[76]
ACC	Astrocytes (GFAP+)		
S1	Astrocytes (GFAP+) Microglia (Iba1+)	Inflammatory/neuropathic pain	[74,162]

Abbreviations: ACC = anterior cingulate cortex; APC = adenomatous polyposis coli tumor suppressor protein; Au1 = primary auditory cortex; DH = dorsal horn of the spinal cord; GFAP = glial fibrillary acidic protein; Iba1 = ionized calcium-binding adaptor molecule 1; OX42 = antibody against integrin αM; S1 = primary somatosensory cortex; SNTN = spinal nucleus of the trigeminal nerve.

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
