# Peer review of "Brain-Derived Neurotrophic Factor, Nociception, and Pain"

_biomolecules, 2024, doi:10.3390/biom14050539_

Round 1

Reviewer 1 Report

Comments and Suggestions for Authors

Author has presented a complete and well documented review article that, under my point of view, shoud be published. I only have some minor corrections and/or suggestions:

Line 75: a double "=" shoud be corrected.

Sentence in lines 114-118 should be shortened.

From page 7 to page 10: citation of figure 1 should indicate figure 2.

Line 257: "subsections" should be added after "following".

Line 476: reference to figure 2 should indicate figure 3.

In vivo and Ex vivo are written either in regular or italic letters. Style should be consistent.

Sentence 534: Sentence should be rewritten to make it clearer. Maybe a comma should be added after "in vivo".

Line 538: add a comma after "whereas" and after "second".

Line 540: add a comma after "rats", delete "thus".

Figure 4 legend: line 907, a ":" should be instead of a "=". The same in line 918.

Table 3: I miss some information regarding oligodendrocytes.

Section 3, first parapragh, last two lines: I think there is a mistake. Otherwise, the sentence should be rewritten to make it clearer.

Author Response

I thank the reviewer for her/his appreciation of my work.

Author has presented a complete and well documented review article that, under my point of view, shoud be published. I only have some minor corrections and/or suggestions:

Line 75: a double "=" shoud be corrected.

Done

Sentence in lines 114-118 should be shortened.

Done

From page 7 to page 10: citation of figure 1 should indicate figure 2.

Changed

Line 257: "subsections" should be added after "following".

Done

Line 476: reference to figure 2 should indicate figure 3.

Done

In vivo and Ex vivo are written either in regular or italic letters. Style should be consistent.

Done

Sentence 534: Sentence should be rewritten to make it clearer. Maybe a comma should be added after "in vivo".

Corrected as follows: Ib1-immunoreactive microglial cells did not stain for BDNF in this study. However, the BDNF released by microglia has been implicated in the generation of neuropathic pain in other in vivo reports.

Line 538: add a comma after "whereas" and after "second".

Done

Line 540: add a comma after "rats", delete "thus".

Done

Figure 4 legend: line 907, a ":" should be instead of a "=". The same in line 918.

Done

Table 3: I miss some information regarding oligodendrocytes.

Info was added

Section 3, first parapragh, last two lines: I think there is a mistake. Otherwise, the sentence should be rewritten to make it clearer.

The last sentences of the paragraph have been rewritten for better clarity.

Reviewer 2 Report

Comments and Suggestions for Authors

The author tried to summarize the roles of BDNG in nociception and pain. However, the manuscript is not prepared, and there are many concerns requiring to be clarified

1. The structure is not clear and hard to read. For example, in the Introduction, the author discussed the synthesis and brief function of BDNF but not related with pain, rather than neurodegenerative diseases. The pain pathway is well know and it is not necessary to state in details.

2. In Figure 1, what is the relationship between LTD/LTP and pain? What is the role of NMDA-R in LTP, it seems in the Figure 1 NMDA-R is not involved in LTP but only in LTD. The cartoon does not include pro-BDNF in the releasing presynaptic vesicle; according to the text, the pro-BDNF domain should bind to the p75NTR after releasing. 

3. In figure 1 legend, only LTD is mentioned but LTP not which is related with sensitization.

4. In the second section (2.1.3), the order should start from Brainstem, Amygdala, Cerebral cortex.

5. Figure 3 is not cited in main text, and what is the evidence that BDNF, SP and CGRP are synthesized, transported and released together in one presynaptic vesicle?

6. The Figure 6 does not show the nociception/pain pathways in non-neuronal cells. 

Abbreviations should be given full names and used consistently after that.

BDNF and mature BDNF should be used consistently

Too many old references

Comments on the Quality of English Language

The language requires to be improved. For example, Line 1394 "However, it is probable that these chemicals also affect peripheral trk receptors in addition to trkB, such as trkA."

Line 281-282, "As reported in Tables 1 and 2 and shown in Figure 1, PSNs located in different ganglia of the PNS express BDNF under basal conditions and/or under circumstances of altered pain perception." There are not Ganglia, should be Figure 2?

Line 50, "BDNF also intervenes in the modulation of pain, the subject of this contribution." what do you mean by this single sentence?

Author Response

The author tried to summarize the roles of BDNG in nociception and pain. However, the manuscript is not prepared, and there are many concerns requiring to be clarified

I am sorry that the MS received such a bad evaluation. I have tried to improve it as much as possible.

  1. The structure is not clear and hard to read. For example, in the Introduction, the author discussed the synthesis and brief function of BDNF but not related with pain, rather than neurodegenerative diseases.

The introduction was written in a very general fashion on purpose as many reviews deal with biochemistry and general biology of BDNF, as well as the intervention of the neurotrophin in neurodegeneration. This is why the subparagraph was entitled “A Brief Overview of BDNF Functions”. To cope with this comment, I have in part rewritten subparagraph 1.1. which is now entitled A Brief Overview of BDNF Functions in the Normal and Pathologic Nervous System and the Cellular Processing of BDNF

The pain pathway is well know and it is not necessary to state in details.

I respectfully disagree with this comment because most of the reviews dealing with the role of BDNF in nociception/pain do not consider the structural/anatomical features whereas I believe that recapitulating these notions is necessary to the comprehension of the other parts of the paper.

  1. In Figure 1, what is the relationship between LTD/LTP and pain? What is the role of NMDA-R in LTP, it seems in the Figure 1 NMDA-R is not involved in LTP but only in LTD. The cartoon does not include pro-BDNF in the releasing presynaptic vesicle; according to the text, the pro-BDNF domain should bind to the p75NTR after releasing.

Thank you very much for these very useful comments on the figure. I have added the NMDA-R to the right side of the cartoon related to LTP. Sorry, but I don’t understand the second comment about the figure: proBDNF (prodomain in red+mature BDNF in blue) is included in the LGVs and binds to p75NTR after release as indicated by the curved arrows.

  1. In figure 1 legend, only LTD is mentioned but LTP not which is related with sensitization.

Thank you again for this comment. The legend has been corrected according to the comment.

  1. In the second section (2.1.3), the order should start from Brainstem, Amygdala, Cerebral cortex.

I have changed the order as indicated.

  1. Figure 3 is not cited in main text, and what is the evidence that BDNF, SP and CGRP are synthesized, transported and released together in one presynaptic vesicle?

The omission of the Figure 3 citation in the main text was a mistake that was corrected accordingly. There is wide evidence that, in neurons, BDNF, CGRP, and SP are synthesized transported, and released together in one presynaptic vesicle. The original paper demonstrating the colocalization of the three bioactive molecules was published by my group in collaboration with the laboratory of Dr. John Priestley (London, UK), see Salio C, Averill S, Priestley JV, Merighi A. Co-storage of BDNF and neuropeptides within individual dense-core vesicles in central and peripheral neurons. Dev Neurobiol. 2007 Feb 15; 67:326-38. In this paper, we have demonstrated by quantitative electron microscopy ICC that in dorsal root ganglion neurons and afferent terminals, and the parabrachial projection to the amygdala, BDNF is stored in individual DCVs with CGRP and substance P. Co-storage occurred in a stoichiometric ratio of 0.7 BDNF:1 CGRP:1 substance P. Remarkably, DCVs contained 31 (spinal cord) -36 (amygdala) times the amount of BDNF detected in agranular vesicles. Subsequent work from my group has further demonstrated this pattern of co-storage in parallel with the co-storage of GDNF, SOM28, and CGRP in the spinal cord dorsal horn, see Salio C, Ferrini F, Muthuraju S, Merighi A. Presynaptic modulation of spinal nociceptive transmission by glial cell line-derived neurotrophic factor (GDNF). J Neurosci. 2014 Oct 8; 34:13819-33. More recently I have quantitatively investigated the size of DCVs in rat DRG neurons and their central terminals in the spinal cord after immunogold labeling for CGRP, neuropeptide K, substance P, neurokinin A or somatostatin to demonstrate the process of fusion between immature vesicles (single-labeled) to give rise to mature vesicles (multiple labeled) in the cell body of these neurons (Merighi A. Costorage of High Molecular Weight Neurotransmitters in Large Dense Core Vesicles of Mammalian Neurons. Front Cell Neurosci. 2018 Aug 21; 12:272.).

These data are (specifically regarding the co-storage of BDNF and substance P) in full accord with the original observations by Michael et al. (G.J Michael, S Averill, A Nitkunan, M Rattray, D.L Bennett, Q Yan, J.V Priestley Nerve growth factor treatment increases brain-derived neurotrophic factor selectively in TrkA-expressing dorsal root ganglion cells and in their central terminations within the spinal cord. J. Neurosci., 17 (1997), pp. 8476-8490) as later discussed by Lessmann, V., Gottmann, K., & Malcangio, M. (2003). Neurotrophin secretion: Current facts and future prospects. Progress in Neurobiology, 69(5), 341-374.

  1. The Figure 6 does not show the nociception/pain pathways in non-neuronal cells.

I believe that this observation refers to Figure 2. Non-neuronal cells have not been shown here because, strictly speaking, there are no non-neuronal pathways. In addition, non-neuronal mechanisms have been considered in Figure 4. To better clarify this in response to the reviewer’s comment, the title of the figure legend has been modified as a Schematic representation of the main neuronal pathways involved in nociception/pain.

Abbreviations should be given full names and used consistently after that.

I have checked these. Thank you.

BDNF and mature BDNF should be used consistently

I have taken care of this as well. I have added a footnote on page 2 to explain the use of mature BDNF in subsection 1.1.

Too many old references

I respectfully disagree with this comment. Of a total of 335 references, 25% are older than 2007 (1st quartile), another 25% are between 2008 and 2015 (2nd quartile/median), the third quartile is at 2020 and the fourth quartile is at 2024. That means that 25% of the references are published in 2020-2024 and 50% in 2015-2024. Apart from this, the oldest references quoted refer to the very first papers published on specific aspects of BDNF biology/function related to pain or neurochemical features of the brain/spinal cord structures of interest in the general frame of the paper. Just to give some examples, the oldest quoted paper, e.g., is a seminal paper on the classification of DRG neurons (ref. 45), and ref. 56 is, to my knowledge the first ultrastructural demonstration of the colocalization of neuropeptides in DRG neurons (1988). Ref. 190 (1991) published in Neuron is the first demonstration of signaling by calcium oscillations in astrocytes. In addition, many of these old papers published before 2001 (refs. 25, 34, 37, 39, 40, 42, 45, 49, 73, 121, 126, 190, 192, 207, 239, 240, 241,) appeared in highly reputed journals such as Science, Nature, J. Neurosci, Neuroscience, J. Comp. Neurol., Pain, Neuron, FEBS Lett., and PNAS.

Comments on the Quality of English Language

The language requires to be improved. For example, Line 1394 "However, it is probable that these chemicals also affect peripheral trk receptors in addition to trkB, such as trkA."

The sentence has been rephrased as follows: However, it seems very likely that, at the periphery, these, chemicals also affect other trk receptors in addition to trkB, such as e.g. trkA.

Line 281-282, "As reported in Tables 1 and 2 and shown in Figure 1, PSNs located in different ganglia of the PNS express BDNF under basal conditions and/or under circumstances of altered pain perception." There are not Ganglia, should be Figure 2?

Sorry, it is indeed Figure 2.

Line 50, "BDNF also intervenes in the modulation of pain, the subject of this contribution." what do you mean by this single sentence?

The sentence has been rewritten.

I have also made an additional check of English language thoroughtout he manuscript.

Round 2

Reviewer 2 Report

Comments and Suggestions for Authors

There are still lots of unrelated statements in the revision. For example, the pain pathway is clear and well-known and it is not necessary to state it in such a length of text, demonstrated by that the manuscript only cite one or none related reference for one paragraph in the 2.3 section.

In contrast to that some opinions are overstated, some discussions are ignored. For example, Line 1354-1355, what are the outcomes of these models? Line 1444-1445, what is the result of the clinical trial?

In Fig 1, the proBDNF and BDNF after releasing from LGV could introduce LTD and LTP, respectively, then what is the net outcome? and under what conditions are they released? the mechanism of LTD and LTP should be discussed in the main text rather than in Fig legend.

In Fig 3, the LTP is induced through BDNF-trkB-PLC-CaMKII-CREB pathway, while the PKC-NMDAR pathway should also be pointed to LTP. And what are the postsynaptic targets of SP and CGRP after they are released together with BDNF from the vesicle? But what is the link between LTP and pain?

In section 2.2.2, only DRG/TG-spinal cord level is discussed, higher level? Even in the spinal cord, what is the role of BDNF in projection neurons, excitatory/inhibitory interneurons?

Why discuss proBDNF in inflammatory pain in vivo and neuropathic pain by BDNF in vivo?

For the references, half of them are published 10 years ago, it is better to discuss publications within 10 years.

Other issues:

Use consistent tense, present or past.

Line 681-682, "...in inflammatory (and 681 neuropathic – see below) pain after...."

Line 1039-1040, "Release of BDNF from microglia is promoted, among others by colony-stimulating 1039 factor 1 (CSF-1)[154,188,189], and/or Wnt3a [190]." A separate sentences does mean anything. 

Comments on the Quality of English Language

The language requires careful read. 

Author Response

There are still lots of unrelated statements in the revision. For example, the pain pathway is clear and well-known and it is not necessary to state it in such a length of text, demonstrated by that the manuscript only cite one or none related reference for one paragraph in the 2.3 section.

As I stated in my previous reply, this information may be useful to readers. To cope with this comment, I have moved it in a Box at the end of the paper and used a smaller font to underline that it can be considered something additional to the main text. Thus readers can decide whether the information is useful to comprehend the main text.

In contrast to that some opinions are overstated, some discussions are ignored. For example, Line 1354-1355, what are the outcomes of these models?

This is now explained in the text as follows: From the studies on these models, it was demonstrated that blocking neuronal or glial BDNF signaling in vitro and/or in vivo impacts the onset of pain-related behaviors and connected cellular signaling.

Line 1444-1445, what is the result of the clinical trial?

The result was partly explained in lines 1445-1450 of the original manuscript. To better clarify them, I have added the following: Mean serum BDNF after treatments with amitriptyline, melatonin, or melatonin + amitriptyline was reduced by 22.57%, 36.6%, and 34.49%, respectively compared to baseline. However, delta values (serum BDNF before treatment minus serum BDNF after treatment) showed a lack of statistically significant differences.

In Fig 1, the proBDNF and BDNF after releasing from LGV could introduce LTD and LTP, respectively, then what is the net outcome? and under what conditions are they released? the mechanism of LTD and LTP should be discussed in the main text rather than in Fig legend.

Thank you for this observation which has been very useful to improve the quality of the paper. In response, I have added a brief description of LTP and LTD mechanisms to the main text expanding the description previously given in the figure legend, which was modified accordingly.

Inhibition of trkB blocks BDNF signaling, whereas blocking p75NTR prevents the signaling of proBDNF. Among the several functions of BDNF or proBDNF upon receptor binding, the regulation of synaptic activity is relevant to the present discussion. It is today widely accepted that mature BDNF can induce long-term potentiation (LTP) whereas proBDNF sustains long-term depression (LTD) at synapses in relation to specific patterns of activity [15].

Sustained by BDNF/trkB, LTP is related to plasticity and subsequent sensitization of the synapse. To induce LTP, BDNF has the potential to directly affect excitatory neurons both pre- and postsynaptically (Figure 2). Additionally, it can alter the balance between excitation and inhibition by inhibiting the GABAergic neurons [16]. For LTP to arise, both the pre-and postsynaptic neurons must be active simultaneously. This is because the postsynaptic neuron needs to be depolarized when glutamate is released from the presynaptic bouton, to completely remove the Mg2+ block of N-methyl-d-aspartate (NMDA) receptors (NMDARs). BDNF-induced LTP involves a series of complex current-voltage relationships of NMDARs and α-amino-3-hydroxy-5-methyl-4-isoxazole propionic acid (AMPA) receptors (AMPARs) [16]. Both receptors are ionotropic and allow the passage of Na+ and K+ ions resulting in a significant influx of Na+ ions and a little efflux of K+ ions, thus causing the postsynaptic neuron to depolarize. When depolarization and glutamate binding occur at the same time, it leads to the maximum influx of calcium through NMDARs. This calcium influx then triggers several intracellular signaling cascades, which ultimately cause changes in synaptic efficiency (Figure 2). NMDARs mostly operate at the postsynaptic membrane, but they have also been observed on presynaptic boutons where they play a role in controlling the release of fast-acting transmitters [17]. The activation of presynaptic receptors can occur through two mechanisms: homosynaptic modulation, which involves significant release from the bouton on which they are situated, and heterosynaptic modulation, which involves the release of glutamate by nearby synapses. The impact of this activation on subsequent release is contingent upon the specific synapse in question [21].

Differently from LTP, LTD can be stimulated by repeated activation of the presynaptic neuron at low frequencies in the absence of postsynaptic activity. Due to the substantial driving force for Ca2+ entry in a resting neuron and the incomplete blockage of NMDARs by Mg2+ even at resting potentials, a significant influx of Ca2+ occurs in response to low-frequency synaptic stimulation. The repeated occurrence of this lower NMDAR-dependent Ca2+ influx likely is what drives the generation of LTD [15].

Since NMDAR-dependent calcium influx triggers both LTP and LTD, the cell needs a mechanism to determine whether to strengthen or weaken a synaptic connection. Today, it is well acknowledged that moderate activation of NMDARs, resulting in a moderate increase in postsynaptic Ca2+, is ideal for initiating LTD. On the other hand, considerably stronger activation of NMDARs, leading to a much bigger increase in postsynaptic calcium, is necessary to cause LTP. Weak activity of the presynaptic neuron leads to modest depolarization and Ca2+influx through NMDA receptors. This activates cellular phosphatases that dephosphorylate AMPARs, thus promoting receptor endocytosis and reinforcing LTD. Remarkably, AMPAR endocytosis can also be sustained by proBDNF after the binding of p75NTR (Figure 1). On the other hand, strong activity matching strong depolarization, as well as mature BDNF binding to trkB, triggers LTP and the subsequent activation of CaMKII, AMPAR phosphorylation and exocytosis (Figure 1).

In Fig 3, the LTP is induced through BDNF-trkB-PLC-CaMKII-CREB pathway, while the PKC-NMDAR pathway should also be pointed to LTP.

Figure 3 (now Figure 2) has been modified accordingly (LTP was added).

And what are the postsynaptic targets of SP and CGRP after they are released together with BDNF from the vesicle?

The discussion of the postsynaptic effects of SP and CGRP at these synapses would be very interesting but I believe it is beyond the purpose of the paper. The original legend of the figure indeed quotes a publication of mine in which the colocalization of the two peptides is discussed. I have expanded it as follows: The two peptides also act as pain modulators and contribute to the excitation of the postsynaptic neurons and nearby neurons in the dorsal horn following volume transmission (see [53] for discussion).

But what is the link between LTP and pain?

I am sorry that the link between LTP and pain was not clear after reading the paper. As discussed in Section 3.2 central sensitization is a form of LTP and this is the link with pain that has been thoroughly discussed in this section of the paper. Yet it may be possible that the link does not show up clearly when one reads the legend of the figure. Thus, I have added the following to the main text and referred to Figure 2 in Section 3.2. Central Sensitization): At PSN/SSN synapses (Figure 2), BDNF enhances glutamate release and modifies the function of AMPARs and NMDARs, resulting in amplified nociceptive signals [80,81] through a mechanism of LTP that is of pivotal importance in the genesis of central sensitization (see Section 3.2. Central Sensitization).

In section 2.2.2, only DRG/TG-spinal cord level is discussed, higher level?

Indeed Section 2.2. is devoted to the description of the Neuronal Mechanisms of BDNF, and there are no functional/mechanistic data on the role of BDNF at higher levels than DRG/spinal cord or TG/SNTN. The studies on high-order neurons that have been quoted in Section 2.1 and Tables 1 and 2 are simply localization studies or studies where upregulation of BDNF was demonstrated following pain induction by heterogenous means - see also Figure 4 (previously Figure 2) in Box 1 that summarizes the results of localization studies relevant to pain modulation. In none of these studies, there is a direct investigation of the modulatory mechanisms sustained by the neurotrophin.

Even in the spinal cord, what is the role of BDNF in projection neurons, excitatory/inhibitory interneurons?

To my knowledge, there is no expression of BDNF in excitatory/inhibitory spinal cord interneurons that may be related to the processing of pain (see my 2018 paper on Progr. Neurobiol. 169:91-134). Likewise, spinal cord projection neurons do not express the neurotrophin (Figure 4).

Why discuss proBDNF in inflammatory pain in vivo and neuropathic pain by BDNF in vivo?

I am sorry but I do not fully understand this observation. Section 2.2.2. (In vivo studies on inflammatory pain modulation by BDNF) was followed by Section 2.2.3. (Modulation of Inflammatory Pain by proBDNF in vivo) because the only available data on the in vivo effects of proBDNF are related to inflammatory pain. I have changed section titles as follows to make things clearer:

2.2.2. (In vivo Studies on Inflammatory Pain Modulation by BDNF and proBDNF)

  • BDNF
  • proBDNF

2.2.3. In vivo Studies on Neuropathic Pain Modulation by BDNF

For the references, half of them are published 10 years ago, it is better to discuss publications within 10 years.

I have answered this observation in my previous rebuttal. This is a personal point of view of the reviewer. There are indeed journals that ask authors to limit the references to be included in a review article to a certain period, but this is not the case for Biomolecules, at least to my knowledge. My point of view is that older references must be quoted if one wants to give credit to those who have conducted prior work. If we hold on to this principle, then there is by no means any reason not to cite old research papers.

Other issues:

Use consistent tense, present or past.

Done

Line 681-682, "...in inflammatory (and 681 neuropathic – see below) pain after...."

Corrected

Line 1039-1040, "Release of BDNF from microglia is promoted, among others by colony-stimulating 1039 factor 1 (CSF-1)[154,188,189], and/or Wnt3a [190]." A separate sentences does mean anything.

I have completed the sentence as follows: …and/or Wnt3a [186]. Notably, CSF-1 mimicked the effects of neuropathic pain in vivo, and its antagonist blockade or gene knockout alleviated neuropathic pain [187], and Wnt3a was downregulated after exercise when neuropathic pain was induced by sciatic nerve damage [188].

Round 3

Reviewer 2 Report

Comments and Suggestions for Authors

none

Author Response

No comments addressed.